# Weight loss practices and eating behaviours among female physique athletes: Acquiring the optimal body composition for competition

Nura Alwan[1,2]☯*, Samantha L. Moss[3]☯, Ian G. Davies[1‡], Kirsty J. Elliott-Sale[4‡], Kevin Enright[1]☯

**1** Research Institute for Sport and Exercise Sciences, Liverpool John Moores University, Liverpool, United Kingdom, **2** Department of Sport and Health Sciences, Oxford Brookes University, Oxford, United Kingdom, **3** Department of Sport and Exercise Sciences, University of Chester, Chester, United Kingdom, **4** Musculoskeletal Physiology Research Group, Sport, Health and Performance Enhancement Research Centre, Nottingham Trent University, Nottingham, United Kingdom

☯ These authors contributed equally to this work.
‡ These authors also contributed equally to this work.
* N.alwan@2016.ljmu.ac.uk

**Data Availability Statement:** Relevant data are within the manuscript. The larger data set cannot be shared publicly because the data contains information which can lead to identifying

## Abstract

Little is known about weight loss practices and eating behaviours in female physique athletes. This study investigated the weight loss history, practices, and key influences during the pre-competition period in a large cohort of female physique athletes stratified by division and experience level. Eating attitudes and behaviours were assessed to identify whether athletes were at risk of developing an eating disorder. Using a cross-sectional research design, female physique athletes ($n$ = 158) were recruited and completed an anonymous online self-reported survey consisting of two validated questionnaires: Rapid Weight Loss Questionnaire and Eating Attitudes Test-26. Irrespective of division or experience, female physique athletes used a combination of weight loss practices during the pre-competition phase. Gradual dieting (94%), food restriction (64%) and excessive exercise (84%), followed by body water manipulation via water loading (73%) were the most commonly used methods. Overall, 37% of female physique athletes were considered at risk of developing an eating disorder. Additionally, 42% of female physique athletes used two pathogenic weight control methods with 34% of Figure novice athletes indicating binge eating once a week or more. The coach (89%) and another athlete (73%) were identified as key influences on athletes' dieting practices and weight loss. The prevalence of athletes identified with disordered eating symptoms and engaging in pathogenic weight control methods is concerning. In future, female physique athletes should seek advice from registered nutritionists to optimise weight management practices and minimise the risk of developing an eating disorder.

## Introduction

Physique athletes are solely judged on aesthetic appearance rather than physical performance [1] and are therefore unique compared to many other aesthetic [2] and weight-sensitive sports [3]. Traditionally, bodybuilding was the only division available for women, however in recent

participants. Considering the small number of physique athletes recruited in some of the divisions, competitive levels and organisations it was deemed necessary to not share the larger data set (outside of the institution or with people outside of the research team) in the ethical application. Data are available from the research ethics committee at Liverpool John Moores [17/TLA/003] (contact via researchethics@livjm.ac.uk) for researchers who meet the criteria for access to confidential data.

**Funding:** The author(s) received no specific funding for this work.

**Competing interests:** The authors have declared that no competing interests exist.

years, further divisions have been introduced to facilitate females with differing physique traits and aspirations [4] including Bikini Fitness and Figure [5]. Female physique athletes (FPA) are not required to be a specific body weight, instead athletes are judged on successfully acquiring a symmetrical and well-proportioned body composition consisting of low-fat mass (FM) and high lean body mass (LBM) [6]. For example, athletes competing in the Figure division are required to have lower levels of FM and greater LBM than the Bikini Fitness division (see [5] for a review of division requirements).

Current practices to acquire optimal body composition for competition are, at present, not well-understood. Consumption of small and frequent meals, combined with rigorous training practices for approximately 11–32 weeks pre-competition have been documented [4,7–10]. However, others show problematic behaviours including extreme dieting, intentional fasting, laxative use, and self-induced vomiting [4,11]. Furthermore, in the final 7-days prior to competition, also known as 'peak week', there is an emphasis on carbohydrate, fluid, and salt manipulation [12–14]. At present, case reports (n = 1) [9,10,15] and studies using larger sample sizes have assessed the weight loss history and dietary practices [16,17] across multiple time-points during a competitive season, however these studies have not captured the frequency of such practices or established other lesser reported practices, which may be used to acquire optimal body composition for competition (*e.g.*, laxatives, water loading and salt manipulation). Delineating the methods used and possible psychological and physical health implications are important, so that FPA can be provided with targeted support that enables safe and effective manipulation of body composition. Recent research in female and male combat sport athletes [18] and powerlifters [19] report coaches to be the primary influence on weight management practices. However, at present, studies investigating key influencers in FPA are limited and should now be explored. Identifying key influencers of practices will offer insights into the reliability of their sources.

Engaging in any dieting or weight loss may put athletes at risk of relative energy deficiency in sport (RED-S) and associated health consequences such as menstrual dysfunction and poor psychological health (for a review see [20]). Repeated engagement with these practices may increase the likelihood of disordered eating (DE) and eating disorders (ED) [21]. Disordered eating begins with voluntary energy restriction, leading to chronic dieting, poor body image, and frequent weight fluctuations using high-risk weight management strategies, ultimately increasing the risk of clinical ED—known as the continuum model of DE [22]. Although it is important to investigate DE and ED in FPA, limited studies have been conducted in physique sports [23] especially with regards to assessing the risk between experience levels and/or divisions [4].

Experience level could influence the likelihood of athletes engaging in aggressive weight loss practices and subsequent DE/ED development [24]. For example, athletes with greater experience may be placed under increased pressure to become leaner from self-expectations and other competitors [25], while others suggest that experienced athletes might better manage expectations [26]. Moreover, the division-specific body composition requirements for competition [5] could also influence strategies for weight management and development of DE/ED. Nevertheless, the influence of both experience level and division remains unexplored.

This study aimed to investigate the weight loss history, practices and influential sources of dieting during the pre-competition period in a large cohort of FPA. A secondary aim was to determine the extent of DE symptoms among FPA, in order to identify whether these athletes were at risk of developing an ED. It was hypothesised that those athletes competing in the Figure division and novice athletes would experience greater weight fluctuations, use acute weight loss practices more frequently and report more DE symptoms compared to Fitness athletes and experienced athletes. Based on previous data from weight-sensitive sports, it was also

hypothesised that coaches would be the main influences on dieting and weight loss practices instead of qualified personnel (irrespective of division and experience).

## Materials and methods

### Recruitment and selection criteria

FPA were recruited from social media advertisement, forums, word-of-mouth, fitness centres and UK-based bodybuilding organisations (June 2017 to July 2018). Inclusion criteria were females aged 18–65 years and participation in a physique competition (previous 12 months). Exclusion criteria included previously diagnosed with an ED, had used, or were currently using, banned substances. By completing the survey, participants provided consent and where informed involvement was voluntary, anonymous and confidential. The study was approved by the University Ethical Review Board in the United Kingdom (17/TLA/003).

### Study design

Using a cross-sectional research design, participants completed an anonymous online self-report survey (Bristol Online Survey© software 2013, Bristol, England) comprising two validated questionnaires (Rapid Weight Loss Questionnaire (RWLQ) [18] and Eating Attitudes Test (EAT-26) [27] to assess weight loss history, practices and DE symptoms, respectively.

Although these questionnaires have not been validated directly in FPA, both questionnaires have been previously utilised in weight-sensitive populations with good sensitivity, specificity and excellent internal consistency (RWLQ: Cronbach's alpha = 0.98) (EAT-26: Cronbach's alpha = 0.90) [18,27]. For constructive validity, the survey was reviewed by two registered practitioners (Sport and Exercise Nutrition Register and Association for Nutrition), and pilot-tested by an experienced international FPA (+5 y competitive experience). Minor syntax, additional words/terminology and formatting modifications were made to the RWLQ questionnaire to ensure appropriateness for physique sports.

### Participants

A total of 191 FPA accessed the survey, of which 178 (93%) completed it. Twenty FPA were excluded (due to not meeting the inclusion criteria and missing data) resulting in 158 FPA in the final analysis and representing the following divisions: Fitness novice (Bikini Fitness and Women's Fitness with ≤1 year of competition experience; $n$ = 62; 39%), Fitness experienced (Bikini Fitness and Women's fitness with >2 years of competition experience; $n$ = 53; 34%), Figure novice (Figure and Physique with ≤1 year of competition experience; $n$ = 19; 12%) and Figure experienced (Figure and Physique with >2 years of competition experience; $n$ = 24; 15%). Previous studies in this population have recruited much smaller cohorts (e.g. n = 26 [11]; n = 14 [28] and hence the recruitment achieved in the current study provided the largest representation to date.

### Rapid weight loss questionnaire

Weight management was assessed using questions derived from the RWLQ, which comprises 21 items and has three subscales: Participant's characteristics (*i.e.,* "*at what age did you start competing*? *and* "*how many times did you compete in your last season*?"), weight and diet history (*i.e.,* "*what is the most weight you have lost*?" *and* "*what is the most weight you have regained in the 7-days after competition*?") and weight loss behaviours (*i.e.,* "*I use gradual dieting*", "*I use excessive exercise*", "*I skip 1 or 2 meals a day*" *and* "*I use water loading*"). FPA indicated agreement using a 5-point Likert Scale: "Never used" (0) and "Always" (3) (with a total

score possible of 54 points) on weight loss behaviours questions. The RWLQ scoring system indicates the higher the score obtained, the more aggressive the weight loss behaviours. This questionnaire was selected because FPAs are known to use weight as a reference for progression and use of acute weight loss practices [11]. The degree of key influential sources of dieting derived from the RWLQ was also assessed ranging from "non-influential" to "very influential".

## Eating Attitude Test-26

Eating attitudes and behaviours were assessed using EAT-26 [27] consisting of three subscales: Dieting (13 items), eating preoccupation (6 items) and oral control (7 items). This 26-item questionnaire asks about pathological eating behaviours and concerns about weight and is a widely used screening instrument for ED risk [29]. Participants rated their agreement with statements such as *"I find myself preoccupied with food"* and *"I am terrified of being overweight"*. Responses were prepared on a 6-point Likert scale anchored by "never" (1), "rarely" (2), "sometimes" (3), "often" (4), "usually" (5) and "always" (6). Additionally, the frequency of pathogenic weight control methods (PWCM) including binge eating, self-induced vomiting and laxatives, diet pills and diuretics (water pills) use was assessed. A total EAT-26 score >20 cut off point is indicative of being at risk for ED.

## Statistical analysis

Statistical Package for the Social Sciences v.23 (SPSS Inc, Chicago, IL) was used for analysis. Continuous data (participant characteristics, weight loss and diet history) was expressed as mean and standard deviations (mean±SD) with ranges, unless otherwise stated, whilst categorical data (frequency and the degree of influence of weight loss practices) were expressed as absolute numbers (n) and percentages (%). Continuous data was checked for the assumption of normality and equality of variances using Kolmogorov-Smirnov and Levene's test, respectively [30]. When normality was met, a two-way between subject's ANOVA compared the variability of mean values across; experience level [26] and division (Fitness and Figure). Bonferroni post-hoc test was used for pairwise comparisons. Where the assumption of normal distribution was violated, the Mann Whitney test was used.

   To assess relationships between the total EAT-26 score and the subscales scores of the EAT-26 and potential risk factors associated with weight history (*i.e.*, weight regain), Spearman's rank correlations ($r_s$) were used. In total, the study recruited from four groups: Bikini Fitness ($n = 107$), Figure ($n = 42$), Women's Fitness ($n = 6$) and Physique athletes ($n = 3$). Due to insufficient respondents from Women's Fitness and Physique athletes, divisions were grouped collectively into "Fitness" (Bikini Fitness and Women's Fitness) and "Figure" (Figure and Physique), owing to the similarities in preparation and proximity in body composition requirements. The significance level was set at $P<0.05$ for all statistical analyses.

## Results

### Participant characteristics

Participant characteristics are presented in Table 1. In total, 99% of respondents reported losing weight for past competitions and participating in 2±1 competitions (range: 1–8) in the previous season. Irrespective of division, novice athletes were younger (27±7 *vs*. 30±7 years, $F_{1,154}$ = 6.73, $P = 0.01$), shorter (163.4±6.4 *vs*. 165.4±6.2 cm, $F_{1,154}$ = 7.74, $P<0.01$) and lighter than experienced athletes (53.8±4.9 *vs*. 56.1±5.8 kg, $F_{1,154}$ = 7.59, $P<0.01$). No main effect of division was identified (P >0.05).

**Table 1. Participant characteristics and weight loss experience for competition across divisions and experience.**

| Athletes (n = 158) | Age (years)* | Height (cm)* | Most recent competition weight (kg)* | Times competed last season (n) | Age at first competition (years) | Typical diet length (weeks) | Competition level % (n) |
|---|---|---|---|---|---|---|---|
| Fitness Novice (n = 62) | 27 ± 6 (18–45) | 164.4 ± 6.5 (153–178.0) | 53.7 ± 5.1 (45–65) | 2 ± 1 (1–6) | 26 ± 7 (17–45) | 15 ± 4 (4–25) | R: 69.4 (43) N:21.0 (13) I:8.1 (5) PRO: 1.6 (1) |
| Fitness Exp (n = 53) | 30 ± 7[a] (20–45) | 165.4 ± 6.2[a] (152–175.3) | 55.3 ± 5.4[a] (44–68) | 2 ± 1 (1–6) | 27 ± 7 (18–43) | 14 ± 6 (2–25) | R: 60.4 (32) N:18.9 (10) I: 11.3 (6) PRO: 9.4 (5) |
| Figure Novice (n = 19) | 29 ± 7 (20–45) | 160.1 ± 5.2 (149.8–170.0) | 54.1 ± 4.2 (44.4–62.0) | 3 ± 2 (1–8) | 28 ± 7 (19–45) | 15 ± 5 (7–32) | R: 63.2 (12) N:31.6 (6) I:5.3 (1) PRO: 0 (0) |
| Figure Exp (n = 24) | 32 ± 6[c] (23–44) | 165.4 ± 6.5[c] (155–178.0) | 57.8 ± 6.5[c] (48.5–70.0) | 3 ± 1 (1–6) | 28 ± 6 (21–42) | 14 ± 3 (10–20) | R: 45.8 (11) N:29.2 (7) I: 20.8 (5) PRO: 4.2 (1) |
| Combined | 29 ± 7 (18–45) | 164.4 ± 6.4 (149.8–178.0) | 54.9 ± 5.5 (44–70) | 2 ± 1 (1–8) | 27 ± 7 (17–45) | 15 ± 5 (2–32) | R: 62 (98) N: 22.8 (36) I: 10.8 (17) PRO 4.4 (7) |

R = regional, N = national, I = International, PRO = professional.

* significant main effect between experience levels.

^ = Typical diet length prior to competition.

R = Regional level, N = National, I = International and PRO = Professional level athlete. Exp = Experienced physique athletes.

[a] denotes significant difference from Fitness Novice, P< 0.05.

[b] denotes significant difference from Fitness Exp, P< 0.05.

[c] denotes significant difference from Figure Novice, P< 0.05.

[d] denotes significant difference from Figure Exp, P< 0.05.

Values are presented as mean ± SD and include the range in brackets.

## Weight loss history

Weight and diet history across divisions and experience levels, is presented in Table 2. Division had a main effect on absolute weight regain in the 7-days after competition (2.7±1.5 *vs.* 2.4±1.7 kg, $F_{1,151}$ = 4.65, $P$ = 0.03) and relative weight regain ($F_{1,149}$ = 8.07, $P<0.01$) with Figure athletes reporting the greatest relative weight regain (5.5%) in the 7-days after competition. No interactions were detected for absolute or relative weight regain in the 7-days after competition. Division and experience level had no effect on usual absolute and relative weight loss ($P>0.05$). There was a division by experience interaction for most weight loss ($F_{1,152}$ = 4.38, $P$ = 0.03) with increases in absolute values from Figure novice athletes (8.4 ±2.4kg) to Figure experienced athletes (10.1±3.6 kg). The opposite was observed, however, between Fitness novice (10.0±4.3 kg) to Fitness experienced (8.8±4.7 kg). There was no main effect for division or experience level in competition week weight loss ($P>0.05$) but a significant interaction ($F_{1,136}$ = 4.46, $P$ = 0.04) was identified. Rapid weight loss score showed no difference between division ($F_{1,153}$ = 1.10, $P$ = 0.30) nor between experience levels ($F_{1,153}$ = 1.10, $P$ = 0.30) and no interaction ($F_{1,153}$ = 0.02, $P$ = 0.90). Relative weight regain was weakly correlated with the total EAT-26 score ($r_s$ = 0.21, $P$ = 0.01), subscales; dieting score ($r_s$ = 0.20, $P$ = 0.01), and bulimia score ($r_s$ = 0.20, $P$ = 0.01).

**Table 2. Female physique athlete (n = 158) responses to weight loss and eating behaviour questions stratified by division and experience level.**

| | Fitness Novice (n = 62) | Fitness Exp (n = 53) | Figure Novice (n = 19) | Figure Exp (n = 24) | Overall (n = 158) |
|---|---|---|---|---|---|
| *Weight usually regained in the week 7-days after competition (kg)? | 2.4 ± 1.7 (0.2–8.5) | 2.3 ± 1.7 (0–8.0) | 3.3 ± 1.6[a] (1.0–6) | 2.3 ± 1.3 [b] (0–5.7) | 2.5 ± 1.7 (0–8.5) |
| *Relative weight usually regained 7-days after competition (%)? | 4.1 ± 3.0 (0–15.9) | 4.0 ± 2.7 (0.7–14.0) | 5.6 ± 2.8[a] (2.0–11.4) | 5.4 ± 3.2[b] (0.7–14.7) | 4.5 ± 3.0 (0–15.9) |
| #*Most weight lost (kg)? | 10.0 ± 4.3 (0–20) | 8.8 ± 4.7[a] (1.5–21.0) | 8.4 ± 2.4[a] (5–15) | 10.1 ± 3.6[bc] (4–20) | 9.4 ± 4.2 (0–21) |
| Most relative weight lost (%)? | 13.7 ± 6.2 (0–35.1) | 15.0 ± 6.1 (6.0–34.4) | 14.9 ± 4.4 (7.0–22.2) | 16.0 ± 6.3 (2.8–28.2) | 14.6 ± 6.0 (0–35.1) |
| Usual weight loss (kg)? | 8.4 ± 3.9 (0–17.5) | 7.6 ± 3.8 (1.5–21.0) | 8.0 ± 1.9 (5.5–12.7) | 8.2 ± 3.7 (0–15) | 8.0 ± 3.6 (0–21) |
| Usual relative weight loss (%)? | 13.0 ± 5.7 (0–25.9) | 11.7 ± 4.8 (2.8–25.3) | 13.0 ± 2.9 (9.7–21.9) | 12.7 ± 5.3 (0–21.1) | 12.5 ± 5.1 (0–25.9) |
| #How much weight do you usually lose in competition week (kg)? | 1.52 ± 1.02 (0–5) | 1.46 ± 0.84[a] (0–4) | 1.70 ± 1.34[a] (0–5) | 1.25 ± 0.68[bc] (0–3) | 1.48 ± 0.96 (0–5) |
| What was your last off-season weight (kg)? | 63.7 ± 6.6 (50.0–82.0) | 63.0 ± 8.3 (49.4–83.0) | 60.9 ± 5.8 (52.2–70) | 64.1 ± 6.2 (52–79.0) | 63.2 ± 7.1 (49.4–83.0) |
| RWLS score | 22.3 ± 7.1 (7.3–48.2) | 21.3 ± 7.3 (10–43.2) | 23.4 ± 7.8 (13–45.7) | 23.2 ± 8.7 (11.5–42) | 22.2 ± 7.5 (7.30–48.20) |
| EAT-26 score | 19.0 ± 12.6 (2.0–54.0) | 17.3 ± 15.2 (0–55) | 22.0 ± 12.7 (2–54) | 18.2 ± 13.1 (1–54) | 18.6 ± 13.5 (0–55) |
| EAT-26 score (≥ 20 cut off) (%) | 38.8 | 36.1 | 42.2 | 21.0 | 36.8 |
| EAT-26 score (< 20 cut off) (%) | 60.5 | 63.2 | 58.0 | 79.4 | 63.5 |
| Dieting score | 11.2 ± 7.9 (0–33) | 11.0 ± 10.2 (0–33.0) | 12.8 ± 8.8 (1–33) | 13.0 ± 13.6 (0–33) | 11.2 ± 8.9 (0–33) |
| Bulimia and food preoccupation score | 4.1 ± 3.8 (0–15) | 3.2 ± 3.5 (0–14) | 5.4 ± 3.6[a] (1–12) | 4.6 ± 4.9[b] (0–13) | 4.1 ± 3.7 (0–15) |
| Oral control score | 3.4 ± 2.8 (0–11) | 2.8 ± 2.8 (0–10) | 3.4 ± 3.1 (0–10) | 4.6 ± 4.2 (0–8) | 3.1 ± 2.8 (0–11) |
| No use of PWCM (%) | 0 | 0 | 15 | 32 | 7.0 |
| Use of 1 PWCM (%) | 41 | 28.5 | 45 | 20 | 37.3 |
| Use of 2 PWCM (%) | 45.9 | 53.8 | 25 | 24 | 42.4 |
| Use of 3 PWCM (%) | 13.1 | 7.7 | 15 | 24 | 13.3 |

*significant main effect between divisions,

#division by experience interaction.

[a] denotes significant difference from Fitness Novice, P< 0.05.

[b] denotes significant difference from Fitness Exp, P< 0.05.

[c] denotes significant difference from Figure Novice, P< 0.05.

[d] denotes significant difference from Figure Exp, P< 0.05.

Exp = Experienced physique athletes. RWLS = Rapid Weight Loss Score, EAT = Eating Attitude Test, PWCM = Pathogenic Weight Control Methods. Most weight loss is the most weight ever cut before a physique competition. Usual weight loss is the weight usually cut before a physique competition. Most relative weight loss is the percentage (of the individual's off-season weight) that was mostly cut for a physique competition. Usual relative weight loss is the percentage (of the individual's off-season weight) that is usually cut for a physique competition.

Values are presented as mean ± SD and include the range in brackets.

## Weight loss methods and sources of influence

Frequency analysis of weight loss practices prior to competition is illustrated in Figs 1 and 2 (*n* = 157). There was no differences between division or experience on any of the weight loss practices; that is, all groups equally practiced similar methods to acquire optimal body compositions (*P* = 0.65). For the combined group 94.0% 'always' or 'sometimes' engaged in gradual dieting (Fig 1A), whilst 84.0% 'always' and 'sometimes' increased exercise levels (Fig 1C) and 64.0% purposely 'always and 'sometimes' restricted foods (Fig 1D). In particular, carbohydrate

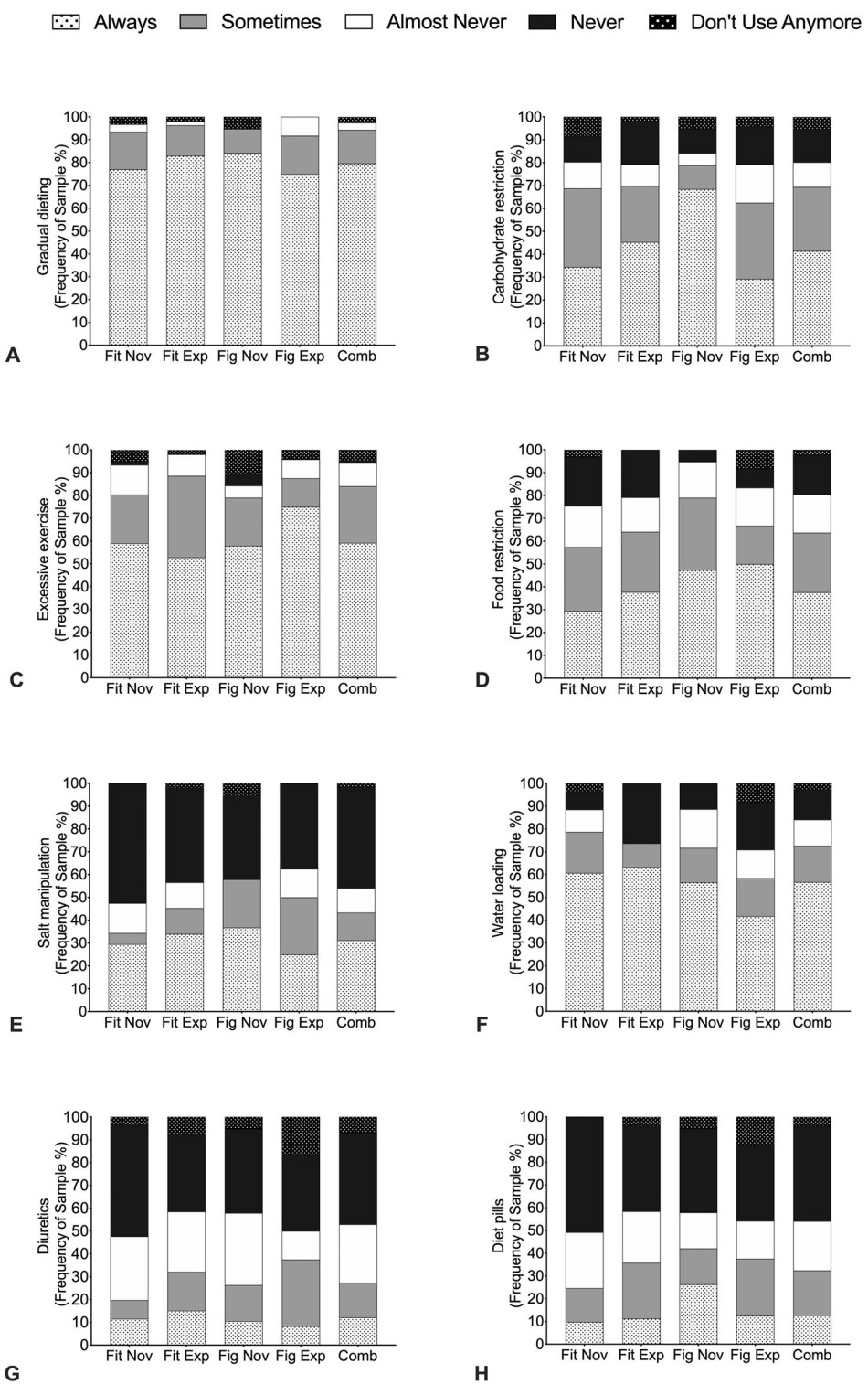

**Fig 1. Frequency analysis of weight loss methods.** A = Gradual dieting B = Carbohydrate restrictions; C = Increasing exercise; D = Food restrictions; E = Salt manipulations, F = Water loading; G = Diuretics and H = Diet pills. Fit Nov = Fitness novice athletes, Fit Exp = Fitness experienced athletes, Fig Nov = Figure novice athletes, Fig Exp = Figure experienced athletes and Comb = Combined female physique athletes.

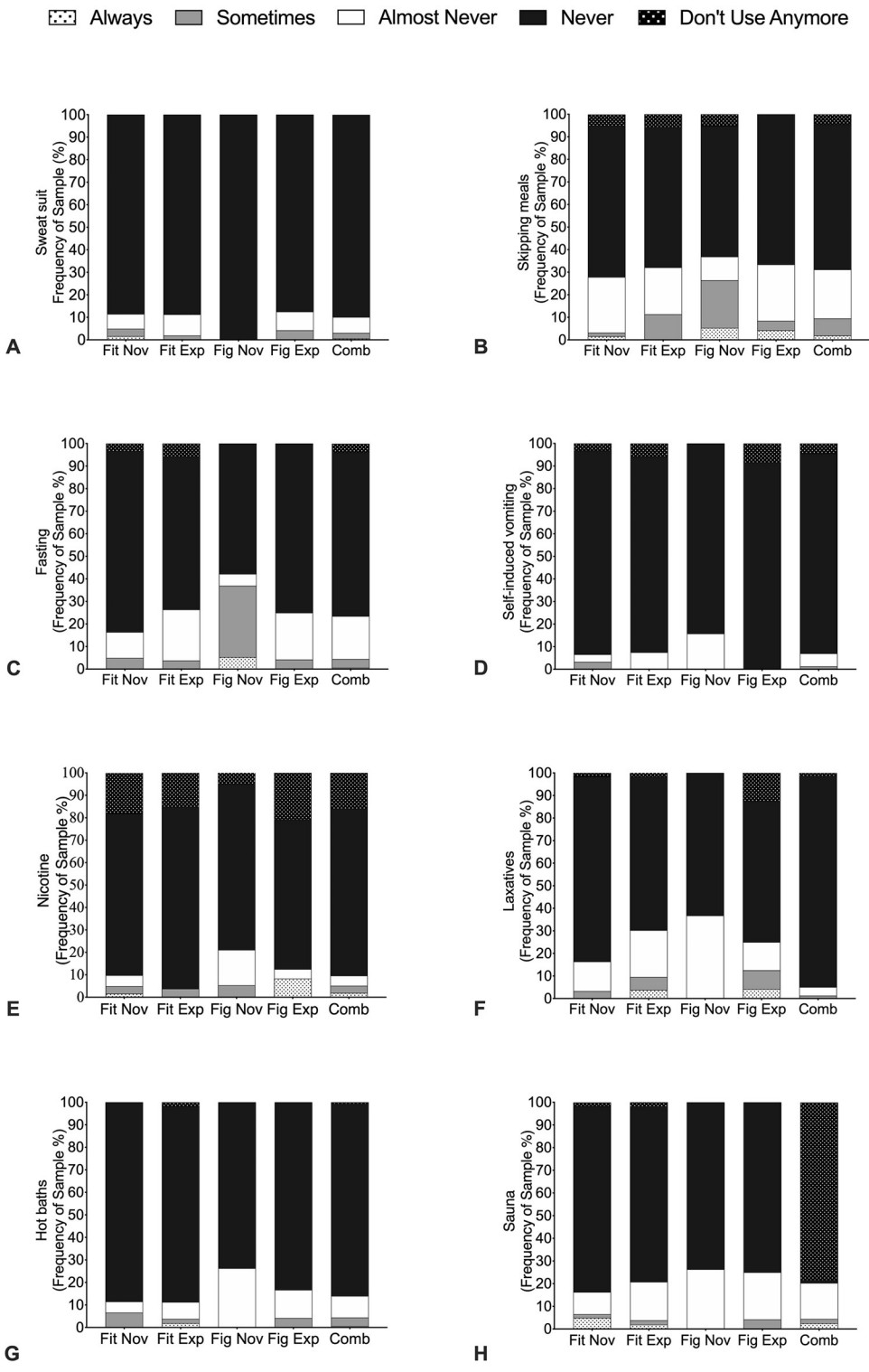

**Fig 2. Frequency analysis of weight loss methods.** A = Sweatsuits; B = Skipping meals; C = Fasting; D = Self-induced vomiting; E = Nicotine; F = Laxatives; G = Hot baths; and H = Sauna. Carbohydrate and fat blockers and Enema were not illustrated here due being used less than 9% by combined groups. Fit Nov = Fitness novice athletes, Fit Exp = Fitness experienced athletes, Fig Nov = Figure novice athletes, Fig Exp = Figure experienced athletes and Comb = Combined female physique athletes.

restrictions were 'always' and 'sometimes' practiced by 70.0% of all athletes (Fig 1B). Water loading in the final pre-competitive period was reported by > 70.0% of the Fitness groups and Figure novice athletes, compared to 58.4% of Figure experienced athletes. Furthermore, 44.0% of all athletes 'regularly' used salt manipulation, with 60.0% of Figure novice athletes 'always' or 'sometimes' using this method prior to competition (Fig 1E). For the combined group skipped meals (86.0%), fasting (91.7%) and laxatives (90.5%) were 'almost never' or 'never used', however Figure novice 'always' or 'sometimes' used skipped meals (26.4%) and fasting (36.9%) (Fig 2). Overall, use of sauna (4.3%) (Fig 2H), sweatsuits (3.1%) (Fig 2A), and hot baths (4.4%) (Fig 2G) were 'always 'or 'sometimes 'used.

The three most influential individuals on dieting and weight loss across all groups were the coach, another physique athlete and partners (Table 3). While the coach had the greatest influence across combined groups with 88.6% of all athletes reporting "quite" and "very influential", a high prevalence reported that parents (59.5%), professional practitioners (82.2%) and nutritionists (52.2%) provide 'no' or 'little' influence on practices.

## Disordered eating symptoms

Nearly 40.0% ($n$ = 56) of participants indicated a score $\geq$ 20 cut-off value and were therefore classified as having DE symptoms and at risk of developing an ED. There was no difference in the total EAT-26 score between novice and experienced athletes (19.2±12.1 *vs.* 17.1±13.9; $P>0.05$). Similarly, there was no difference in the total EAT-26 score between Fitness and Figure athletes (18.8±13.9 *vs.* 16.5±10.2; $P>0.05$). Oral control and dieting sub-scale scores from the EAT-26 test showed no differences between division or experience ($P>0.05$). There was a main effect, however, between divisions on bulimia and food preoccupation score where Figure athletes scored greater than Fitness athletes, irrespective of the experience (3.8±3.7 *vs.* 4.7±3.7; $P$ = 0.01).

## Pathogenic weight control methods

The magnitude of PWCM, as a measure to control weight (Fig 3), were assessed using the behavioural section of the EAT-26 [27]. For the combined group, 42.4% used two out of three

**Table 3. Frequency analysis of the persons who are influential on the dieting practices and weight loss of female physique athletes (n = 158) stratified by division and experience level.**

| Persons | Not influential % | | | | | A little influential % | | | | | Unsure % | | | | | Quite influential % | | | | | Very influential % | | | | |
|---|---|---|---|---|---|---|---|---|---|---|---|---|---|---|---|---|---|---|---|---|---|---|---|---|---|
| | Fit Nov | Fit Exp | Fig Nov | Fig Exp | C | Fit Nov | Fit Exp | Fig Nov | Fig Exp | C | Fit Nov | Fit Exp | Fig Nov | Fig Exp | C | Fit Nov | Fit Exp | Fig Nov | Fig Exp | C | Fit Nov | Fit Exp | Fig Nov | Fig Exp | C |
| Coach | 8.1 | 15.1 | 0 | 4.2 | 8.9 | 0 | 3.8 | 0 | 0 | 1.3 | 0 | 1.9 | 5.3 | 0 | 1.3 | 21.0 | 20.8 | 15.8 | 16.7 | 19.6 | 71.0 | 58.5 | 78.9 | 79.2 | 69.0 |
| Doctor | 77.4 | 82.7 | 94.7 | 83.3 | 82.2 | 11.3 | 9.6 | 0 | 8.3 | 8.9 | 6.5 | 5.8 | 5.3 | 8.3 | 6.4 | 1.6 | 0 | 0 | 0 | 0.6 | 1.9 | 2.6 | 0 | 0 | 1.9 |
| Internet | 30.6 | 30.2 | 42.1 | 25.0 | 31.0 | 12.9 | 30.2 | 10.5 | 20.8 | 19.6 | 8.1 | 3.8 | 5.3 | 12.5 | 7.0 | 29 | 24.5 | 21.1 | 37.5 | 27.8 | 19.4 | 11.3 | 21.1 | 4.2 | 14.6 |
| Another athlete | 12.9 | 11.3 | 10.5 | 4.2 | 10.8 | 12.9 | 13.2 | 10.5 | 12.5 | 12.7 | 1.6 | 1.9 | 15.8 | 0 | 3.2 | 45.2 | 45.3 | 36.8 | 45.8 | 44.3 | 27.4 | 28.3 | 26.3 | 37.5 | 29.1 |
| Friends | 38.7 | 39.6 | 68.4 | 33.3 | 41.8 | 29.0 | 26.4 | 15.8 | 29.2 | 26.6 | 4.8 | 5.7 | 0 | 4.2 | 4.4 | 17.7 | 22.6 | 0 | 20.8 | 17.7 | 9.7 | 5.7 | 15.8 | 12.5 | 9.5 |
| Partner | 25.8 | 22.6 | 47.4 | 16.7 | 25.9 | 12.9 | 24.5 | 0 | 16.7 | 15.8 | 6.5 | 9.4 | 0 | 4.2 | 6.3 | 21.0 | 30.2 | 15.8 | 25.0 | 24.1 | 33.9 | 13.2 | 36.8 | 37.5 | 27.8 |
| Parents | 64.5 | 56.6 | 68.4 | 45.8 | 59.5 | 12.9 | 18.9 | 15.8 | 25.0 | 17.1 | 1.6 | 3.8 | 0 | 4.2 | 2.5 | 16.1 | 15.1 | 10.5 | 16.7 | 15.2 | 4.8 | 5.7 | 5.3 | 8.3 | 5.7 |
| Training partner | 37.1 | 45.3 | 52.6 | 33.3 | 41.1 | 16.1 | 20.8 | 10.5 | 25.0 | 18.4 | 8.1 | 7.5 | 0 | 0 | 5.7 | 25.8 | 18.9 | 26.3 | 20.8 | 22.8 | 12.9 | 7.5 | 0 | 20.8 | 12.0 |
| Nutritionist/ dietitian | 45.9 | 45.3 | 84.2 | 58.3 | 52.2 | 13.1 | 5.7 | 0 | 8.3 | 8.3 | 9.8 | 11.3 | 0 | 0 | 7.6 | 13.1 | 18.9 | 0 | 16.7 | 14.0 | 18.0 | 18.9 | 15.8 | 16.7 | 17.8 |

Fit Nov = Fitness novice athletes, Fit Exp = Fitness experienced athletes, Fig Nov = Figure novice athletes, Fig Exp = Figure experienced athletes and C = Combined female physique athletes.

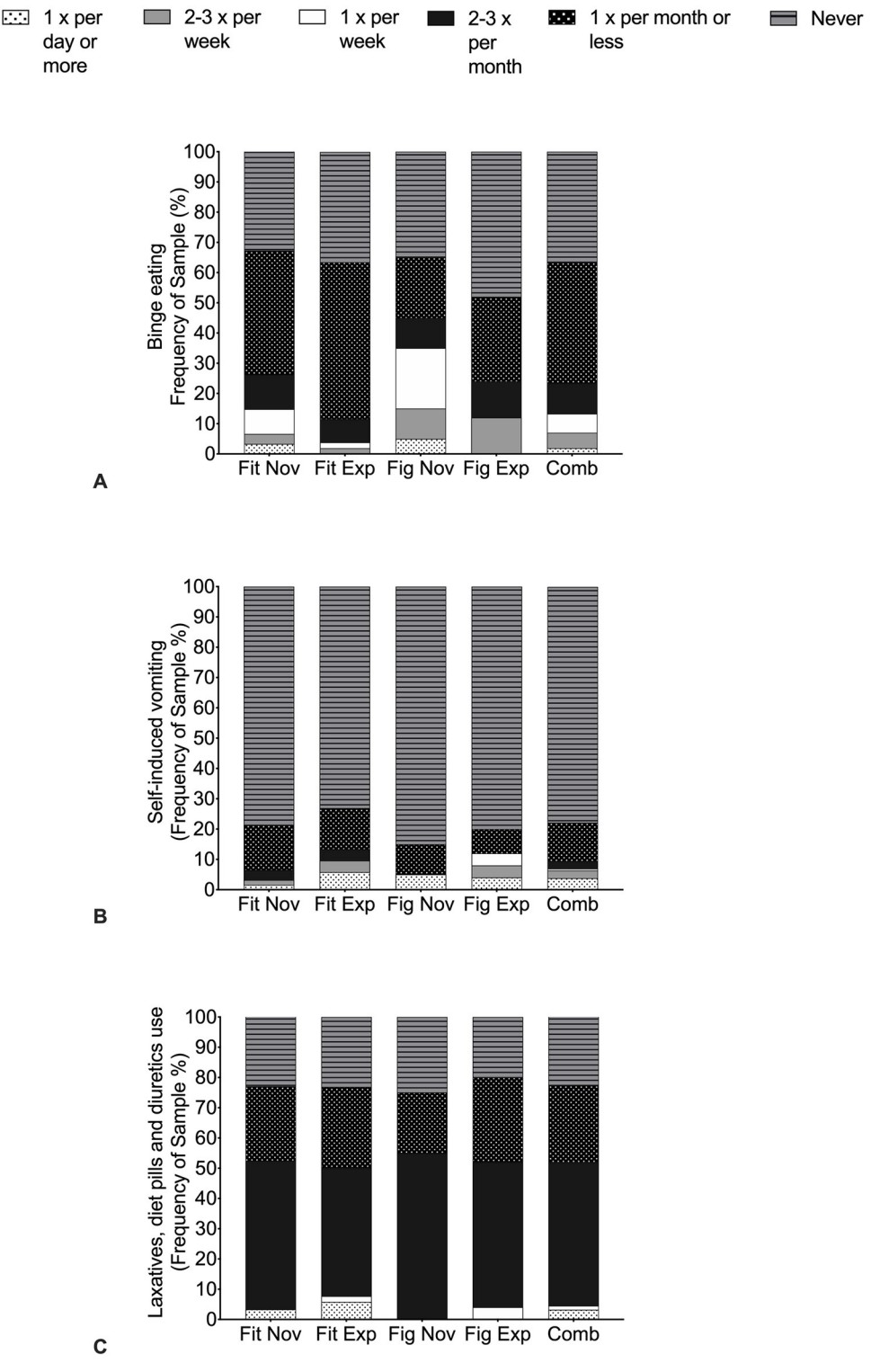

**Fig 3. Frequency analysis of pathogenic weight control methods used by physique athletes from the Eating Attitude Test-26 questionnaire.** A = Binge Eating; B = Self-induced vomiting; C = Laxatives, diet pills and diuretics (water pills) use. Fit Nov = Fitness novice athletes, Fit Exp = Fitness experienced athletes, Fig Nov = Figure novice athletes, Fig Exp = Figure experienced athletes and Comb = Combined female physique athletes.

PWCM (laxatives, diet pills and diuretics, and binge eating), while 13.3% used all methods to manage their weight. Laxatives, diet pills and diuretics were the most popular methods to manage weight amongst all FPA (Fig 2C) with a large percentage of Fitness novice (77.1%), Fitness experienced (76.9%), Figure novice (77.0%) and Figure experienced (80.0%) athletes reporting their use once a month or more. Likewise, binge eating occurred once a month or more for 63.3% of all FPA. Of those, 14.8% of Fitness novice, 3.8% Fitness experienced, 35.0% of Figure novice and 0.0% Figure experienced athletes indicated binge eating once a week or more (Fig 3A). Self-induced vomiting (Fig 3B) occurred once a month or more in 21.3%, 26.9%, 15.0% and 20.0% of Fitness novice, Fitness experienced, Figure novice and Figure experienced athletes, respectively.

## Discussion

This study provides data that assesses the weight loss history and frequency of practices in FPA alongside influential sources of dieting and DE symptoms. The results show that irrespective of division or experience level, FPA use gradual dieting, restricting foods, excessive exercise and acute body water manipulation weight loss practices in pre-competition which provides additional knowledge on dieting procedures among this group. Coaches and other athletes were identified as the key influencers of dieting and weight loss practices. Furthermore, 37% of FPA were considered at risk of developing an ED. Lastly, almost half of the athletes used two PWCMs with over a third of Figure novice athletes reporting binge eating once a week or more.

In the present study, almost all (99%) FPA (n = 157) reported losing weight pre-competition, which was similar to previous studies assessing weight loss history [9,10,16,17]; although, the magnitude of weight loss varied. In particular, Fitness novice athletes (10 ± 4.3 kg) lost more weight (in absolute values) than Fitness experienced athletes (8.8 ± 4.7 kg), with the opposite observed for Figure athletes (experienced = 10.1 ± 3.6 kg; novice = 8.4 ± 2.4 kg). Anecdotally, in their first year of competing, Fitness novice athletes present with more FM than Fitness experienced athletes and therefore lose more weight prior to competition. However, this remains speculative and warrants future research. Furthermore, Figure experienced athletes may place greater emphasis on increasing LBM during the off-season (and therefore also may experience concomitant increases in their levels of FM) compared to a Figure novice athlete [15] which subsequently requires greater weight loss as they enter the pre-competition phase. A number of athletes reported regaining significant body weight, with Figure athletes reporting the greatest relative weight regain (5.5%) in the 7-days after competition compared to Fitness athletes (4.1%). Interestingly though, the greatest weight individual regain was reported by a Fitness novice athlete, who increased body mass by 15.9% (8.5 kg) in the first 7-days post-competition indicating that individual variability within groups was apparent. A reasonable explanation for the greater relative weight regain in Figure athletes may be due to the higher frequency of binge eating self-reported by Figure athletes (47%) compared to the Fitness athletes (18.6%) and the greater score of bulimia and food preoccupation in Figure athletes compared to Fitness athletes observed in the present study [31]. Our findings confirm previous observations in FPA [32,33], which show similar rapid changes in body mass post-competition. Rapid changes to post-competition body mass could be explained by rebound hyperphagia, which often occurs in response to chronic and intensive periods of energy restriction [34,35] and may increase the risk for obesity and cardiometabolic diseases in later life [36].

In preparation for competition, FPA implement gradual dieting, food restrictions, excessive exercise and water loading. In attempting to lose weight, restrictive eating and excessive

exercise is implemented and could potentially affect metabolic [32] and endocrine health such as menstrual dysfunction [9] thus increasing the risk of RED-S [17]. Our findings also confirm and extend recent work from Chappell and Simper (2018) [12] showing that a high proportion of FPA will 'always' or 'sometimes' use carbohydrate, salt, and fluid manipulation in the final 7-days pre-competition. However, it is worth noting that Chappell and Simper (2018) [12] did not separate athletes into respective divisions, which could be important because of the different body composition requirements [5]. Water loading (*i.e.*, consuming approximately ~10 litres of water per day for 3–5 days), followed by reduced water intake each day leading into the competition with complete fluid restriction over the 10–24 hours before the performance on the stage [37], was used by >70% of the FPA in this study. Despite the high prevalence, water loading in isolation has been suggested to be a safe and effective method to increase urine production for a maximal fluid driven weight loss in combat sports [38]. However, this practice when combined with diuretics and electrolyte manipulation presents additional dangers including hyponatremia, which has resulted in a number of fatalities [39,40] and is not a recommended method to regulate weight [41]. A unique finding of our study was the higher prevalence of diet pills and diuretics (32 and 27%, respectively) to lose body water compared to previous work, which did not report these methods [11]. A possible explanation for the disparity between studies could be due to our larger sample size (*n* = 26 versus *n* = 158, respectively) and the addition of new divisions in recent years which require specific body composition requirements [5].

FPA reported coaches and other physique athletes to be the greatest influences on their dieting and weight loss efforts, whilst medical doctors and nutritionists were not seen as influential (Table 3). This agrees with other studies on weight-sensitive sports [4,7,18]. For example, Chappell et al., (2018) [7] reported that 14 out of 16 FPA used a coach for guidance with training, dietary practices and feedback on their physique. A possible explanation for why coaches appear to be highly influential is that many former athletes become coaches toward the end of their careers [37]. This might result in systemic cultural attitudes toward aggressive weight loss practices being preserved within physique sports, although, at present, this is anecdotal and warrants further investigation. If this is the case, collectively these data highlight the need for coaches to be adequately educated regarding safe and effective methods of weight management in an attempt to improve the health and well-being of athletes. Bodybuilding and fitness organisations should consider providing continuing professional development courses from suitable professionals for members and encourage the support of multidisciplinary teams (*i.e.*, registered sports psychologists and sports nutritionists/dietitians). In addition, online resources (41%) were 'quite' and 'very' influential methods of information for athletes. Seeking information from internet resources has previously been shown to be a significant predictor of ED [42,43] and therefore highlights the importance of educating athletes correctly, as unregulated online advice could be detrimental to training, performance and health.

Our data show that 37% of the combined group scored greater than the cut-off value for the EAT-26 questionnaire. The proportion of FPA considered at risk of ED in this study strongly agrees with Whitehead et al., (2020) (using a comparable EDI questionnaire) which showed that 50% of Australian FPA were identified with DE [4] and Money-Taylor et al., (2021) [44] (using a similar screening tool to that of the present study) which showed that 57% of international FPA were identified with DE symptoms. Furthermore, the overall total EAT-26 score of 19 with Figure novice athletes scoring higher than the other groups (22 vs. 17–19). This finding suggests Figure novice athletes are at greater risk for ED than Figure experienced, and Fitness groups. Due to the specific demands of the Figure division (*i.e.*, lower FM and higher FFM than the Fitness division), novice athletes beginning Figure competitions may be exposed to

body composition requirements that are more challenging to achieve [45]. It is also possible that women may be susceptible or already exhibit DE symptoms, as they enter physique sports, in hope of alleviating these symptoms. As such, it is speculative whether long-term engagement in the sport provides a protective effect from ED risks in FPA [46]. Although, EAT-26 questionnaire has been validated in other athletic groups, further work to validate it in this specific population is necessary.

For PWCM, over 40% percent of all athletes engaged with laxatives, diet pills and diuretics, and binge eating to manage weight, whilst 13% of athletes reported engaging with all PWCM. These percentages are similar to those reported previously on FPA [4,47]. FPA commonly experience pressures to remain extremely lean [25], which can lead to cycling between restrictive eating behaviours and excessive exercise, to repeated binge eating episodes [23]. This is of particular interest considering FPA repeat weight cycling episodes and therefore may experience higher rates of obesity and cardiometabolic disease later in life [36]. A recent observation suggests that athletes who have an underlying predisposition to developing an ED are more attracted to participate in physique sports [37]. For example, key psychopathological traits underpinning the development of DE are often seen in physique sports [24,26]. Our data showed high levels of food restraint amongst all participants (approximately 40% reported 'always' food restricting), which is in agreement with a recent report [23] who revealed higher eating restraint in FPA compared to healthy controls before, during and post dieting. Moreover, Mathisen and Sundgot-Borgen (2019) [23] reported that 2 (6%) and 9 (28%) out of 33 Scandinavian FPA (at the time of the study) had currently or have previously been diagnosed with an ED diagnosed by health personnel. These findings have serious implications for FPA and those managing these athletes such as coaches with respect to detecting DE early. Thus, education programmes should encourage organisations, coaches and physique athletes themselves to seek support from multidisciplinary professionals on the identification, prevention and management of DE/ED, and this should apply to all female athletes adhering to the extreme modern body ideal emphasising a lean and muscular aesthetic look.

Although the overall sample size in our study (n = 158) was substantially larger than previous cross-sectional studies in this population (n = 26 [11]; n = 14 [28], the small number of participants competing in Women's Fitness and Physique divisions meant that it was not possible to analyse the differences between every division. However, it must be acknowledged that athletes competing in Women's Fitness, Wellness and Physique divisions are more likely to engage with prohibited substances given the expectations to achieve less FM and greater LBM [48–50]. Moreover, RWLQ used in this study was primarily developed for combat sports athletes, and thus may lack specificity for aesthetic sports. Therefore, this screening tool needs to be validated in physique sports and to complement non-validated questions for the exploration of novel practices (*i.e.*, low-fibre diets). Additionally, athletes gave self-reported responses and therefore may have over or under-reported weight loss and DE behaviours [51]. Although EAT-26 has been validated in athletic groups and widely used in physique and bodybuilding populations [24,52], on its own it does not yield a specific diagnosis of an ED. For a diagnosis to occur, a follow up clinical interview should be performed with those who score ≥20, however, this was beyond the scope of this study. Finally, although the EAT-26 questionnaire contains components which could be interpreted as pathological eating behaviours (*i.e.*, *'I engage in dieting behaviour'* and *'Aware of the calorie content of foods that I eat'*), these could well be normal behaviours which are required to compete in physique sports, rather than being indicative of DE per se. As such, distinguishing between true pathological and non-pathological behaviours eating behaviours should be carefully considered when interpretating the DE symptoms results.

In an attempt to further understand the weight management of FPA and potential risks of acute and chronic health outcomes, future work should i) validate EAT-26 questionnaire in physique athletes, ii) assess dietary intake using weighed food inventory and food diaries, alongside training practices (modality, training volume and loads) during the pre-competition phase and iii) collect blood samples and other measurements for analysis of markers relating to metabolic, renal and endocrine health (especially on changes in competition week). Using qualitative techniques to characterise the physical and psychological health implications when participating in physique sports (*i.e.*, lived experiences) could also further understanding.

## Conclusion

Irrespective of division or experience, FPA used various weight loss practices including gradual dieting and acute body water manipulation in the pre-competition phase. Figure novice athletes reported most weight loss pre-competition, alongside significant increases in weight gain post-competition, indicating patterns of weight cycling. Coaches and other athletes had most influence on athletes' practices. Finally, 37% of FPA were at risk of developing ED with 42% using two PWCM throughout the season. Therefore, we encourage FPA and coaches to utilise appropriate expertise (*e.g.*, registered nutritionists and psychologists) when preparing for competition. This might reduce the risks associated with severe weight loss practices evident in this population.

## Acknowledgments

We thank Professor Kevin Lamb at University of Chester for his guidance with the preparation of the manuscript. We also thank all the FPA for providing their time to complete this survey.

## Author Contributions

**Conceptualization:** Nura Alwan, Samantha L. Moss, Ian G. Davies, Kevin Enright.

**Data curation:** Nura Alwan, Samantha L. Moss, Kevin Enright.

**Formal analysis:** Nura Alwan, Samantha L. Moss.

**Investigation:** Nura Alwan, Kevin Enright.

**Methodology:** Nura Alwan, Ian G. Davies, Kevin Enright.

**Project administration:** Nura Alwan, Kevin Enright.

**Software:** Nura Alwan.

**Supervision:** Samantha L. Moss, Ian G. Davies, Kirsty J. Elliott-Sale, Kevin Enright.

**Validation:** Ian G. Davies.

**Writing – original draft:** Nura Alwan.

**Writing – review & editing:** Nura Alwan, Samantha L. Moss, Ian G. Davies, Kirsty J. Elliott-Sale, Kevin Enright.

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
