## [Decision Letter · Decision Letter 0]

18 Jun 2021

PONE-D-21-15998

Weight loss practices and eating behaviours among female physique athletes: Acquiring the optimal body composition for competition

PLOS ONE

Dear Dr. Alwan,

Thank you for submitting your manuscript to PLOS ONE. After careful consideration, we feel that it has merit but does not fully meet PLOS ONE’s publication criteria as it currently stands. Therefore, we invite you to submit a revised version of the manuscript that addresses the points raised during the review process.

Although the work is considered to be interesting, several substantive points have been raised during the review process. Please carefully consider the comments provided as they indicate areas that need to be refined.  

We look forward to receiving your revised manuscript.

Kind regards,

Cherilyn N. McLester, PhD

Academic Editor

PLOS ONE

Journal Requirements:

Reviewers' comments:

Reviewer's Responses to Questions

**Comments to the Author**

1. Is the manuscript technically sound, and do the data support the conclusions?

Reviewer #1: Partly

Reviewer #2: Yes

2. Has the statistical analysis been performed appropriately and rigorously? 

Reviewer #1: Yes

Reviewer #2: Yes

3. Have the authors made all data underlying the findings in their manuscript fully available?

Reviewer #1: No

Reviewer #2: Yes

4. Is the manuscript presented in an intelligible fashion and written in standard English?

Reviewer #1: Yes

Reviewer #2: Yes

5. Review Comments to the Author

Reviewer #1: The work is interesting, considering the popularity of this sport, and it short existence – hence limited publications on health effects. Still – there are a few detailed publications already, which seems not too familiar to the current authors considering the poor refers to this previous work. Hence, my main feedback relates to the vague knowledge of previous findings, or at least incomplete use of references. Also, considering the increased number of publications studying physique athletes, references to bodybuilders should be de-emphasized. E.g.; Line 55-56: here many more publications should be referred to; Mathisen et al 2019; Helms et al 2019; Hulmi et al 2017; Trexler et al 2017; Petrizzo et al 2017; Halliday et al 2016. Acknowledging these authors work, and the current knowledge, these should be referred to more often throughout the manuscript, and also brought up for comparison in the discussion (rather than claiming no one else has studies the phenomena at quest). As many of these already is in the reference list, it seems the authors have not read the work done – this applies specifically to the request for future studies in line 131-137!

The strength of the current study is the additional knowledge it provides on dieting procedures, which should be the main focus; adding to the details we now know about health effects from participation in this sport.

Abstract: the authors conclude that all used evidence-based methods for dieting, however this is not presented or discussed in the manuscript -neither do I find any reason to claim this, considering the results presented.

Introduction

Line 60; weight loss history has been presented previously; e.g. Mathisen et al 2019.

Line 62; both mental and physical health implications are important.

Line 69-70; while the quote is not wrong, it is more accurate to say that all situations leading to low energy availability will possibly result in RED-s – as such, any dieting attempt will be a high-risk scenario.

Line 73; I assume the authors means DE (disordered eating) in the beginning of the sentence.

Line 76, and 89-91; most of the previous mentioned work has evaluated DE and ED, and even selfreported EDs diagnosed by health personnel (Mathisen et al 2019).

Line 100-1001; exclusion criteria is an ED, but the study aim to study the frequency of risk for EDs? As previous studies have found rather high prevalence of EDs in this athlete group, there will be a rather biased sample if exclusion criterion is a diagnosis of ED.

Do the authors have any hypothesis? Specifically explaining the rationale to compare athletes according to disciplines and to experience!

Methods

line 122: definition for the different experience-categories needs to be clarified.

RWLQ needs a proper presentation; e.g. how many items, tha Chronbachs-alpha in the current sample. The latter also relates to the presentation of the EAT-26.

EAT-26 also should be presented with Chronbachs-alpha for the current sample.

Line 165: Do the author mean they will study relations between a total score and a subscale-score of the EAT-26?

It would be interesting (and relevant) to see the different physique categories represented in the sample; only two are mentioned, being those with few participants, hence grouped to two other overall categories)

General remark regarding tables; please indicate which results are significant different to each other, not the variables name.

Table 1: tha variable “typical diet length” misses unit of measurement

Line 186-187: Althoug presented in table 2, the period of time which the regain refers to, should be mentioned in the text

Table 2: There is a need for more thourugh explanation of the variables; what are usual weight lost and what is most weight lost (or for what time frame?); what is relative weight lost?

Line 216-219: please consider rephrasing, as the content is not easy to understand. More than 80% never used the methods, still the majority of novice figure athletes did?

Any information on the competence of the coaches? (as these are found most influential on dieting practces?!)

Discussion:

(this is not the first report on DE or ED!)

Line 21: “as a result” does not fit into context here.

Figure athletes had the largest relative weight regain – discuss implications, and please add time-perspectives when presenting weight regain results.

Line 35-38: while the claims raised regarding the risk of RED-s is correct, the authors needs to acknowledge that the main reason for risk of RED-s in their athletes is attempts to lose weight, and the fact that they did lose weight (i.e. low energy availability)

Line 50-52 (and lines leading up to this): considering the harm by using weight-reduction methods relying on electrolyte balances, the authors should emphasise the potential health threat, rather than encourage more studies to fine a “safe method”. This sort of method is not recommended as a weight regulation practice (e.g. see publications by the IOC expert group on REDs).

Line 60; please refer the authors of the publication directly in the text

Line 85-89; the literature repeatedly reveal that females are more prone to ED than males, hence, a comparison to male bodybuilders seems not relevant. There are more reports on ED or atheltes at ED-risk in previously mentioned and recommended references! To follow up on this; the line 90-94 therefor needs to be re-evaluated - e.g. Mathisen et al 2019 reported numbers with previous or current ED diagnosed by health personnel) Line 105-109; here too, Mathisen et al 2019 revealed higher eating restrains compared to control group before, throughout and after dieting period.’

Line 131-137: most of the requested information for future studies, are already presented in previous publications; e.g. weighed food records, blood samples, techniques to characterize physical changes and body composition; please see previous listed references

One finding deserves more focus; Novice fitness athletes lost more weight than experienced ones; opposite within figure athletes. Discuss possible reasons to this observation.

The conclusion would do better with a shorter summary and implications.

Reviewer #2: Thank you to the authors for submitting their paper titled Weight loss practices and eating behaviours among female physique athletes: Acquiring the optimal body composition for competition. Herein they describe their cross-sectional study examining weight loss behaviours and disordered eating symptoms in a large sample of competitive female physique athletes, as well as sources used to obtain weight loss practices. Results showed a high percentage of participants demonstrated increased disordered eating behaviours via the EAT-26 and a variety of weight loss behaviours through the RWLQ. Overall the paper is well structured, clear, and mostly concise, presenting a well developed background and justification for the study, clear methods and results.

Apart from some minor suggestions presented below, my major comment is in regards to the interpretation and discussion of findings. The nature of physique competition preparation requires athletes to undertake many behaviours and practices which are interpreted as pathological in eating disorder questionnaires (in this case the EAT-26) when in fact they are just inherent components of competition preparation. Examples include knowing the calorie content of food I eat (item 6), particularly avoid high carbohydrate foods (item 7), think about burning up calories when I exercise (item 11), display self-control around food (item 19), and engage in dieting behaviour (item 23). Distinguishing these inherent components from true pathological qualities is a challenge in this line of research. As such, I feel a more nuanced, balanced approach to the discussion is required.

In addition to this, the inclusion criteria indicate participants resistance trained at least 5 times per week, which is a very high training frequency. I am concerned that such a criteria would exclude a large number of FPA which would introduce a selection bias. There may also be an argument that this high frequency criteria may lead to a biased recruitment of athletes who over-exercise and are at risk of disordered eating/exercise behaviours.

Specific comments are as follows:

Abstract

Line 37 - The use of evidence-based practices by participants was not highlighted in the manuscript, so I feel it would not be appropriate to include here in the abstract.

Introduction

Lines 45-52 - It may be beneficial to the reader if you were able to provide more context to the different division of competition. What distinguishes the different divisions (Body composition/leanness? Muscularity? Other?). Given your examination aims to compare based on division, this information would be useful for readers.

Line 111 - It is common to report the Cronbach's alpha of the included questionnaires. As the RWLQ was adapted for the population under investigation, I would suggest this measure of internal consistency be reported.

Line 129 - please include the number of items in the RWLQ. It would also be helpful if an idea of the participant characteristic questions are given (section 1). Detail around what kind of weight loss behaviours are included in the questionnaire would give the reader more context. Admittedly this is presented in the results, however for clarity having that detail here is useful.

Results:

Table 1 - Typical diet length (column 7): presumably this is measured in weeks. I suggest this should be indicated for clarity.

Table 1 legend - * indicates main effect for division. Please clarify this, as the text in the previous paragraph (line 174-179) indicates no effect of division which if I am reading correctly contradicts the table.

Weight loss history (beginning line 185) - Unclear why weight regain is presented first here in the text, but not first in table 2. I would suggest consistency in presentation order makes the results more reader-friendly.

Figures - Is it possible to amend the y-axes labels on the figures to include the weight loss method? For example, Figure 1A "gradual dieting frequency (%)". I feel this would make the figures easier to read without having to go back and forth to the figure legends.

Discussion

Line 82 - Argument regarding inexperience places athletes at greater risk of DE. Could this argument be expanded? Why would inexperience increase risk for DE/ED? Is there some protective effect in remaining in the sport, longevity helps alleviate symptoms/risk? Perhaps the lower risk in more experienced participants suggests those at risk do not remain in the sport? A similar argument is briefly presented regarding muscle dysmorphia symptoms in bodybuilders in Mitchell et al (2017) http://dx.doi.org/10.1016/j.bodyim.2017.04.003

Lines 79-114 - following from my main comment above, this section in particular requires some thought or at least balance regarding the limitations of the EAT-26 in distinguishing pathological behaviours from inherent components of competition preparation. Line 106 regarding food restraint, perhaps food restraint is simply a requirement to continually achieve fat loss in preparation for a show.

Limitations paragraph (lines 116-130) - I think worth including the limitation surrounding the EAT of distinguishing pathological from non-pathological behaviours in this particular population. Perhaps a suggestion for future research could be to validate the EAT-26 in physique athletes?

6. PLOS authors have the option to publish the peer review history of their article (what does this mean?). If published, this will include your full peer review and any attached files.

Reviewer #1: No

Reviewer #2: **Yes: **Lachlan Mitchell

---

## [Author Response · Author response to Decision Letter 0]

5 Aug 2021

PONE-D-21-15998

Weight loss practices and eating behaviours among female physique athletes: Acquiring the optimal body composition for competition

PLOS ONE

Dear Dr. Alwan,

Thank you for submitting your manuscript to PLOS ONE. After careful consideration, we feel that it has merit but does not fully meet PLOS ONE’s publication criteria as it currently stands. Therefore, we invite you to submit a revised version of the manuscript that addresses the points raised during the review process.

Although the work is considered to be interesting, several substantive points have been raised during the review process. Please carefully consider the comments provided as they indicate areas that need to be refined. 

We look forward to receiving your revised manuscript.

Kind regards,

Cherilyn N. McLester, PhD

Academic Editor

PLOS ONE

https://hes32-ctp.trendmicro.com:443/wis/clicktime/v1/query?url=https%3a%2f%2fjournals.plos.org%2fplosone%2fs%2ffile%3fid%3dwjVg%2fPLOSOne%5fformatting%5fsample%5fmain%5fbody.pdf&umid=fea4fa62-af1d-4bd7-b784-0a3d48fdda92&auth=e7d0e2218e6161c78f47b51039a961804123edc5-0b8cfd7b6088dc7561ad14f8f7411c427828233c

and 

https://hes32-ctp.trendmicro.com:443/wis/clicktime/v1/query?url=https%3a%2f%2fjournals.plos.org%2fplosone%2fs%2ffile%3fid%3dba62%2fPLOSOne%5fformatting%5fsample%5ftitle%5fauthors%5faffiliations.pdf&umid=fea4fa62-af1d-4bd7-b784-0a3d48fdda92&auth=e7d0e2218e6161c78f47b51039a961804123edc5-071f926254a8c03911f562a237d20c16a42221fa

Thank you for signposting us to this information, we have now made these formatting changes.

2. We note that you have indicated that data from this study are available upon request. PLOS only allows data to be available upon request if there are legal or ethical restrictions on sharing data publicly. For more information on unacceptable data access restrictions. Please see https://hes32-ctp.trendmicro.com:443/wis/clicktime/v1/query?url=http%3a%2f%2fjournals.plos.org%2fplosone%2fs%2fdata%2davailability%23loc%2dunacceptable%2ddata%2daccess%2drestrictions&umid=fea4fa62-af1d-4bd7-b784-0a3d48fdda92&auth=e7d0e2218e6161c78f47b51039a961804123edc5-f4fa0ec81795f11bc391f284f9163e678cb2cb54.

The data contains information which can lead to identifying of participants. Considering the small number of physique athletes recruited in some of the divisions, competitive levels and organisations it was deemed necessary to not share the larger data set (outside of the institution or with people outside of the research team) in the ethical application. The details of research ethics committee are Liverpool John Moores [17/TLA/003]

Response to reviewer comments

Reviewer 1:

The work is interesting, considering the popularity of this sport, and it short existence – hence limited publications on health effects. Still – there are a few detailed publications already, which seems not too familiar to the current authors considering the poor refers to this previous work. Hence, my main feedback relates to the vague knowledge of previous findings, or at least incomplete use of references. Also, considering the increased number of publications studying physique athletes, references to bodybuilders should be de-emphasized. E.g.; Line 55-56: here many more publications should be referred to; Mathisen et al 2019; Helms et al 2019; Hulmi et al 2017; Trexler et al 2017; Petrizzo et al 2017; Halliday et al 2016. Acknowledging these authors work, and the current knowledge, these should be referred to more often throughout the manuscript, and also brought up for comparison in the discussion (rather than claiming no one else has studies the phenomena at quest). As many of these already is in the reference list, it seems the authors have not read the work done – this applies specifically to the request for future studies in line 131-137! The strength of the current study is the additional knowledge it provides on dieting procedures, which should be the main focus; adding to the details we now know about health effects from participation in this sport.

Thank you for your positive and constructive comments. We have included and developed further the suggested references throughout the manuscript, which we hope has provided a more balanced overview of the key findings in this area. These now feature more consistently throughout the manuscript. For example, we have added more relevant studies (on female physique athletes) between lines 59-60 (Whitehead et al., 2020, Chappell et al., 2018, Rohrig et al., 2017, Halliday et al., 2016, and Petrizzo et al., 2017) and 63-64 in the introduction (Halliday et al. 2016, Petrizzo et al., 2017, Tinsley et al., 2019, Hulmi et al., 2016, and Mathisen et al., 2019) and line 310 in the discussion (Halliday et al., 2016, Petrizzo et al., 2017, Hulmi et al., 2016, and Mathisen et al., 2019) to reduce the emphasis on bodybuilders. Furthermore, we have added the reference Mathisen and Sundgot-Borgen (2019) for comparisons with regard to eating restraint in the discussion between the lines 407-409. Lastly, we have edited lines 440-444 of future research directions and tried to highlight better the novelty of our work in the manuscript.

Abstract

The authors conclude that all used evidence-based methods for dieting, however this is not presented or discussed in the manuscript -neither do I find any reason to claim this, considering the results presented.

We thank the reviewer for highlighting this point. We have now changed this to ‘acute and chronic weight loss practices’ on line 38 in the abstract. Additionally in the manuscript we also present it as acute and chronic on line 232-233 in the results and line 301-302 in the discussion. We feel that we have described the practices they are undertaking e.g., water loading (acute) and gradual dieting (chronic) and hope this now reflects the results. 

Introduction

Line 60; weight loss history has been presented previously; e.g. Mathisen et al 2019.

We thank the reviewer for this suggestion. We have now inserted into the manuscript studies by Hulmi et al., 2016 and Mathisen et al., 2019 and other case reports by Petrizzo et al., 2017, Haliday et al., 206, and Tinsley et al., 2019. The manuscript now reads as follows between lines 63-68 in the introduction: 

“At present, case reports (n=1) [12,14] and studies using larger sample sizes have assessed the weight loss history and dietary practices [14,15] across multiple time-points during a competitive season, however these studies have not captured the frequency of such practices or established other lesser reported practices, which may be used to acquire optimal body composition for competition (e.g., laxatives, water loading, and salt manipulation).”

Line 62; both mental and physical health implications are important.

Thank you for making this suggestion. We have made the following change on line 68 in the introduction: 

“Delineating the methods used and possible psychological and physical health implications are important,….”

Line 69-70; while the quote is not wrong, it is more accurate to say that all situations leading to low energy availability will possibly result in RED-s – as such, any dieting attempt will be a high-risk scenario.

We thank the reviewer for this suggestion and we agree with this comment. This has now been replaced (on line 75 in the introduction) with: 

‘Engaging in any dieting or weight loss may put athletes at risk of relative energy deficiency in sport…….’.

Line 73; I assume the authors means DE (disordered eating) in the beginning of the sentence.

We apologise for this error. This is now corrected (on line 79 in the introduction) to: 

“Disordered eating…”. 

Line 76, and 89-91; most of the previous mentioned work has evaluated DE and ED, and even self reported EDs diagnosed by health personnel (Mathisen et al 2019).

We thank the reviewer for highlighting this point. We have now included the study by Mathisen et al., 2019 on line 83. We acknowledge previous work has been done on the prevalence of ED and DE, however, we were attempting to highlight that, limited studies have been conducted in the area and this is especially the case when comparing the different divisions and experience level. We understand how this may not have been clear and as such the sentence (on lines 82-84 in the introduction) is rewritten to the following: 

“…..limited studies have been conducted in physique sports ‘[22], especially with regards to assessing the risk between experience levels and/or divisions [4].’

Hopefully this has will be clearer to the reader. 

Line 100-1001; exclusion criteria is an ED, but the study aim to study the frequency of risk for EDs? As previous studies have found rather high prevalence of EDs in this athlete group, there will be a rather biased sample if exclusion criterion is a diagnosis of ED.

We thank the reviewer for this valuable comment. We would like to highlight that the aim of this study was to assess the extent/prevalence of DE symptoms and not the frequency of risk for ED. Please see line 95 in the introduction for the study aim. Regarding the decision to exclude participants with an eating disorder, we were advised by the University ethics committee to exclude participants previously diagnosed ED’s. Furthermore, we believe that recruiting athletes who may not have recognised that they have DE symptoms will add to the current knowledge, and by doing so investigate a previously unreported issue in the sport

Do the authors have any hypothesis? Specifically explaining the rationale to compare athletes according to disciplines and to experience!

Thank you for this question. We agree that including a hypothesis will be a positive addition to the manuscript and this is now included between lines 96 to 101 in the introduction: 

“It was hypothesised that those athletes competing in the Figure division and novice athletes would experience greater weight fluctuations, use acute weight loss practices more frequently and report more DE symptoms compared to Fitness athletes and experienced athletes. Based on previous data from weight-sensitive sports, it was also hypothesised that coaches would be the main influences on dieting and weight loss practices instead of qualified personnel (irrespective of division and experience).”

We have attempted to explain the rationale in the introduction and so to avoid unnecessary repetition and streamline the writing, we hope this is considered appropriate. For example, between lines 85-92 we have attempted to explain the rationale why exploring difference between experience levels and between divisions is important. 

Methods

line 122: definition for the different experience-categories needs to be clarified.

A definition of experience levels was originally included in the statistical analysis section; however, we agree this will be better suited within the participants section in the methods. This section (between line 133-137 in the participant section of the methods) now reads: 

“……representing the following divisions: Fitness novice (Bikini Fitness and Women’s Fitness with ≤1 year of competition experience; n = 62; 39%), Fitness experienced (Bikini Fitness and Women’s Fitness with >2 years of competition experience; n = 53; 34%), Figure novice (Figure and Physique with ≤1 year of competition experience; n = 19; 12%) and Figure experienced (Figure and Physique with >2 years of competition experience; n = 24; 15%).”

RWLQ needs a proper presentation; e.g. how many items, tha Chronbachs-alpha in the current sample. The latter also relates to the presentation of the EAT-26.

The amount of items of the RWLQ is now included in the RWLQ section of the methods on line 143: 

“….from the RWLQ, which comprises of 21 items and has three subscales:…”

We have now included the Cronbach’s-alpha on line 122 in the Study design of the methods and this now reads: 

“Both questionnaires have been previously utilised in aesthetic and weight-sensitive populations with good sensitivity, specificity and excellent internal consistency (RWLQ: Cronbach's alpha = 0.98) (EAT-26: Cronbach's alpha = 0.90) [18,27].”

EAT-26 also should be presented with Chronbachs-alpha for the current sample.

We thank the reviewer for this suggestion. In line with the previous comment we have included this on line 122 in the Study design of the methods section. 

Line 165: Do the author mean they will study relations between a total score and a subscale-score of the EAT-26?

We agree that this sentence is not clear. We have now replaced (line 179-180) with: 

“…. relationships between the total EAT-26 score and the subscales scores of the EAT-26 and potential risk factors associated with weight history (i.e., weight regain)”.

It would be interesting (and relevant) to see the different physique categories represented in the sample; only two are mentioned, being those with few participants, hence grouped to two other overall categories)

We thank the reviewer for this suggestion. We agree that this could be an interesting insight into the population recruited and we have therefore included (between line 181-182 in the statistical analysis section) the following statement: 

“In total, the study recruited from four groups: Bikini Fitness (n=107), Figure (n=42), Women’s Fitness (n=6) and Physique Athletes (n=3)”.

Results

General remark regarding tables; please indicate which results are significant different to each other, not the variables name.

We thank the reviewer for highlighting this and we have now added alphabetic symbols for significance on the results as well as adding an asterisk or hashtags to the variables in Table 1 and 2 on line 196 and 220-221.

Table 1: the variable “typical diet length” misses unit of measurement

have now added “(weeks)” into column 7 in Table 1 on line 196. 

Line 186-187: Although presented in table 2, the period of time which the regain refers to, should be mentioned in the text

We have referred to this earlier in the manuscript in the section Rapid weight loss questionnaire section on line 145-146. We agree that ‘weight regain in the 7-days after competition’ could be repeated in the results and is therefore inserted in on line 204, 206-207 and line 207-208 in the results section. 

Table 2: There is a need for more thorough explanation of the variables; what are usual weight lost and what is most weight lost (or for what time frame?); what is relative weight lost?

Thank you for this suggestion. We have now added a section below Table 2 explaining (relative) usual weight loss and (relative) most weight loss in more detail, to allow the reader to understand the results better. This now reads: 

“Most weight loss is the most weight ever cut before a physique competition. Usual weight loss is the weight usually cut before a physique competition. Most relative weight loss is the percentage (of the individual’s off-season weight) that was mostly cut for a physique competition. Usual relative weight loss is the percentage (of the individual’s off-season weight) that is usually cut for a physique competition.” 

Line 216-219: please consider rephrasing, as the content is not easy to understand. More than 80% never used the methods, still the majority of novice figure athletes did? Any information on the competence of the coaches? (as these are found most influential on dieting practces?!)

We thank the reviewer for raising this comment. We have rephrased (on lines 242-244) this to:

“For the combined group skipped meals (86.0%), fasting (91.7%), and laxatives (90.5%) were ‘almost never’ or ‘never used’, however Figure novice ‘always’ or ‘sometimes’ used skipped meals (26.4%) and fasting (36.9%)…”.

Unfortunately, we do not have any data on coaches because we did not recruit coaches. 

Discussion:

(this is not the first report on DE or ED!)

We apologise for this omission and have now changed (see line 299-300 in the discussion) it this section to: 

“This study provides data that assesses the weight loss history and frequency of practices in FPA alongside influential sources of dieting and DE symptoms”. 

Line 21: “as a result” does not fit into context here.

We have changed this to “Moreover” on line 305 in the discussion.

One finding deserves more focus; Novice fitness athletes lost more weight than experienced ones; opposite within figure athletes. Discuss possible reasons to this observation.

Thank you for this suggestion. We have changed this section and it now reads (between lines 313-319): 

“Anecdotally, in their first year of competing, Fitness novice athletes present with more FM than Fitness experienced athletes and therefore lose more weight prior to competition. However, this remains speculative and warrants future research. Furthermore, Figure experienced athletes may place greater emphasis on increasing LBM during the off-season (and therefore also may experience concomitant increases in their levels of FM) compared to a Figure novice athlete, which subsequently requires greater weight loss as they enter the pre-competition phase.”

Figure athletes had the largest relative weight regain – discuss implications, and please add time-perspectives when presenting weight regain results.

We thank the reviewer for highlighting this comment. The time frame has now been added on line 321 and 323.

We have also highlighted the potential reason why relative weight regain in the 7-days after competition is greater in figure than fitness athletes and lines 324-328 in discussion now read as: “A reasonable explanation for the greater relative weight regain in Figure athletes may be due the higher frequency of binge eating self-reported by Figure athletes (47%) compared to the Fitness athletes (18.6%) and the greater score of bulimia and food preoccupation in Figure athletes compared to Fitness athletes observed in the present study"

In order to provide implications of the effects of weight cycling on health we have added (on line 332-333): “…and may increase the risk for obesity and cardiometabolic diseases in later life [34]”

Line 35-38: while the claims raised regarding the risk of RED-s is correct, the authors needs to acknowledge that the main reason for risk of RED-s in their athletes is attempts to lose weight, and the fact that they did lose weight (i.e. low energy availability)

Thank you for this suggestion. We agree this was not clear and this sentence (lines 334-337 in the discussion) has now been restructured to read:

“In preparation for competition, FPA implement gradual dieting, food restrictions, excessive exercise and water loading. In attempting to lose weight, restrictive eating and excessive exercise is implemented and could potentially affect metabolic [32] and endocrine health such as menstrual dysfunction [9] thus increasing the risk of RED-S [17].” 

Line 50-52 (and lines leading up to this): considering the harm by using weight-reduction methods relying on electrolyte balances, the authors should emphasise the potential health threat, rather than encourage more studies to fine a “safe method”. This sort of method is not recommended as a weight regulation practice (e.g. see publications by the IOC expert group on REDs).

We thank the reviewer for this remark. We agree and the sentence has therefore been removed on line 348-350. We have also added the reference from IOC expert group to emphasise the danger of using such methods. This reads as following: 

“However, this practice when combined with diuretics and electrolyte manipulation presents additional dangers including hyponatremia ,which has resulted in a number of fatalities [39,40] and is not a recommended method to regulate weight [41]”.

Line 60; please refer the authors of the publication directly in the text

We have now added the authors name directly in the sentence line 359 in the discussion so it reads; 

“This agrees with other studies on weight-sensitive sports [4,14,37]. For example, Chappell et al., (2018) [7] reported….”.

Line 85-89; the literature repeatedly reveal that females are more prone to ED than males, hence, a comparison to male bodybuilders seems not relevant. There are more reports on ED or atheltes at ED-risk in previously mentioned and recommended references! To follow up on this; the line 90-94 therefor needs to be re-evaluated - e.g. Mathisen et al 2019 reported numbers with previous or current ED diagnosed by health personnel) Line 105-109; here too, Mathisen et al 2019 revealed higher eating restrains compared to control group before, throughout and after dieting period.’

We thank the reviewer for raising this valuable point. Although we are aware many female studies have been investigated using different tools of measurement of DE and ED, we believe that it is appropriate to also compare studies using similar tools as explained on line 386. We agree that the Mathisen study is an important reference to use in this discussion and we have now added this into manuscript in two different places (line 406-412): 

“Our data showed high levels of food restraint amongst all participants (approximately 40% reported ‘always’ food restricting), which is in agreement with a recent report [23] who revealed higher eating restraints in FPA compared to healthy controls before, during and post dieting”. 

“Moreover, Mathisen and Sundgot-Borgen (2019) [23] reported that 9 (28%) and 2 (6%) out of 33 Scandinavian FPA (at the time of the study) had currently or have previously been diagnosed with an ED diagnosed by health personnel.”

Line 131-137: most of the requested information for future studies, are already presented in previous publications; e.g. weighed food records, blood samples, techniques to characterize physical changes and body composition; please see previous listed references

We thank the reviewer for making this remark and we would like to clarify further. 

The second future research suggestion we suggested was aimed at assessing dietary practices alongside training practices. This sentence has now been updated to enhance clarity:

“…assess dietary intake using weighed food inventory and food diaries alongside training practices (modality, training volume and loads) during the pre-competition...”.

Whilst we are aware there are many studies that have assessed dietary intake in the pre-competition phase including Halliday et al., 2016, Mathisen et al., 2019, Petrizzo et al., 2017, Tinsley et al., 2019 and Longstrom et al., 2021, we are also aware that limited studies have completed this assessment alongside training practices. Documentation of training practices in FP athletes is mostly reliant on metabolic equivalent data (Halliday et al., 2016; Petrizzo et al., 2017; Hulmi et al., 2016), which provides estimates of exercise energy expenditure, but does detail the specific training they perform. It is therefore unknown if/how training variables are manipulated by female physique athletes who are actively engaging in weight loss in order to achieve their body composition aspirations for competition. 

With regard to the third research suggestion. We have now amended the sentence, so it reads:

“…collect blood samples and other measurements for analysis of markers relating to metabolic, renal, and endocrine health (especially on changes in competition week)”.

Whilst previous work has assessed some blood/saliva markers in their studies such as Hulmi et al., 2016 (Endocrine markers and metabolic markers), Trexler et al., 2017 (Endocrine and appetite hormones) and Rohrig et al., 2017 (vitamins, metabolic and endocrine markers) our suggestion is to provide a comprehensive profile of markers that indicate metabolic, renal, and endocrine status, with particular reference to competition week. 

Hulmi has taken measurements of metabolic markers (e.g., T3 and TSH) and endocrine markers (e.g., oestradiol and testosterone), at three time points, in different phases of the competitive season (baseline, pre-competition and post-competition). Trexler took measurements of endocrine hormones (e.g., insulin and testosterone using saliva) just before competition and post-competition, and Rohrig assessed a small profile of endocrine, lipid and metabolic markers every month leading into competition. As such, we believe that it is necessary for more studies to assess clinical blood markers in FPA as this could contribute to our understanding of RED-S and renal function. We are also aware that no study to date has assessed changes in the competition week, which is likely to be a crucial window given the dietary manipulation such as carbohydrate restriction and loading, salt and fluid manipulation etc. as shown by Chappell et al., (2017). 

The conclusion would do better with a shorter summary and implications.

We thank the reviewer for this suggestion; this has now been refined and now reads: 

“Irrespective of division or experience, FPA used a combination of acute and chronic weight loss practices pre-competition. Figure novice athletes reported most weight loss pre-competition, alongside significant increases in weight gain post-competition, indicating patterns of weight cycling. Coaches and other athletes had most influence on athletes’ practices. Finally, 37% of FPA were at risk of developing ED with 42% using two PWCM throughout the season. Therefore, we encourage FPA and coaches to utilise appropriate expertise (e.g., registered nutritionists and psychologists) when preparing for competition. This might reduce the risks associated with severe weight loss practices evident in this population. “

Reviewer #2: 

Thank you to the authors for submitting their paper titled Weight loss practices and eating behaviours among female physique athletes: Acquiring the optimal body composition for competition. Herein they describe their cross-sectional study examining weight loss behaviours and disordered eating symptoms in a large sample of competitive female physique athletes, as well as sources used to obtain weight loss practices. Results showed a high percentage of participants demonstrated increased disordered eating behaviours via the EAT-26 and a variety of weight loss behaviours through the RWLQ. Overall the paper is well structured, clear, and mostly concise, presenting a well developed background and justification for the study, clear methods and results.

Thank you for these positive comments. 

Apart from some minor suggestions presented below, my major comment is in regards to the interpretation and discussion of findings. The nature of physique competition preparation requires athletes to undertake many behaviours and practices which are interpreted as pathological in eating disorder questionnaires (in this case the EAT-26) when in fact they are just inherent components of competition preparation. Examples include knowing the calorie content of food I eat (item 6), particularly avoid high carbohydrate foods (item 7), think about burning up calories when I exercise (item 11), display self-control around food (item 19), and engage in dieting behaviour (item 23). Distinguishing these inherent components from true pathological qualities is a challenge in this line of research. As such, I feel a more nuanced, balanced approach to the discussion is required.

We thank the reviewer for this valuable and fair comment. We have reflected on this and believe it is important to include in the discussion of this manuscript. We have therefore considered this in the manuscript between lines 433-439 and reads:

“Finally, although the EAT-26 questionnaire contains components which could be interpretated as pathological eating behaviours (i.e., ‘I engage in dieting behaviour’ and ‘Aware of the calorie content of foods that I eat’), these could well be normal behaviours which are required to compete in physique sports, rather than being indicative of an DE per se. As such, distinguishing between true pathological and non-pathological behaviours eating behaviours should be carefully considered when interpretating the DE symptoms results.”

In addition to this, the inclusion criteria indicate participants resistance trained at least 5 times per week, which is a very high training frequency. I am concerned that such a criteria would exclude a large number of FPA which would introduce a selection bias. There may also be an argument that this high frequency criteria may lead to a biased recruitment of athletes who over-exercise and are at risk of disordered eating/exercise behaviours.

We apologise, we accidentally misreported our inclusion criteria. We have now amended this in the manuscript, and it now reads:

“Inclusion criteria were females aged 18-65 years and participation in a physique competition (previous 12 months).” 

We have included this extract from our ethics application to show our actual inclusion criteria: “ C4a. What are the inclusion criteria? (Please include information on how you will ensure that your participants will be informed of your inclusion criteria and how you will ensure that any specific inclusion criteria are met) 

“Participants will be females between 18 and 65 years of age, and have been competed previously on stage in the last two competitive seasons. Participants will receive this information verbally, on recruitment poster and/or on the participant information form when finding out about the study. When conducting the study, the demographic information will include a question on age and competing history so that any participants not meeting the inclusion criteria can be removed from the study. If they meet the inclusion criteria, an option “Take me to the survey” will be clicked, and participant can continue to the survey. “[17/TLA/003 Liverpool John Moores University Ethics committee] “

Line 37 - The use of evidence-based practices by participants was not highlighted in the manuscript, so I feel it would not be appropriate to include here in the abstract.

Thank you for this comment. We have now changed this to “acute and chronic weight loss practices” on line 38 in the abstract. We feel that we have described the practices they are undertaking e.g., water loading (acute) and gradual dieting (chronic) and hope this now reflects the results. 

Introduction

Lines 45-52 - It may be beneficial to the reader if you were able to provide more context to the different division of competition. What distinguishes the different divisions (Body composition/leanness? Muscularity? Other?). Given your examination aims to compare based on division, this information would be useful for readers.

We have now added a sentence which reads: 

“For example, athletes competing in the Figure division are required to have lower levels of FM and greater LBM than the Bikini Fitness division (see [5] for a review of division requirements)”.We hope this provides some useful information for readers, whilst not adding excessively to the word limit imposed by the journal. 

Line 111 - It is common to report the Cronbach's alpha of the included questionnaires. As the RWLQ was adapted for the population under investigation, I would suggest this measure of internal consistency be reported.

We thank the reviewer for highlighting this point. This is now included on line 122 and states: “…excellent internal consistency (RWLQ: Cronbach's alpha = 0.98) (EAT-26: Cronbach's alpha = 0.90) [18,27]”.

Line 129 - please include the number of items in the RWLQ. It would also be helpful if an idea of the participant characteristic questions are given (section 1). Detail around what kind of weight loss behaviours are included in the questionnaire would give the reader more context. Admittedly this is presented in the results, however for clarity having that detail here is useful.

We thank the reviewer for making this suggestion. This now reads (line 143-144): 

“…which comprises 21 items and has three subscales: Participant’s characteristics (i.e., “at what age did you start competing? and “how many times did you compete in your last season?”)”

Results:

Table 1 - Typical diet length (column 7): presumably this is measured in weeks. I suggest this should be indicated for clarity.

We thank the reviewer for spotting this missing unit. We have now added the unit “(weeks)” to Table 1 in column 7 on line 196.

Table 1 legend - * indicates main effect for division. Please clarify this, as the text in the previous paragraph (line 174-179) indicates no effect of division which if I am reading correctly contradicts the table.

We thank the reviewer for highlighting this error. Below Table 1, we have now replaced ‘division’ with ‘experience’ reflecting our results provided in participant characteristics section. 

Weight loss history (beginning line 185) - Unclear why weight regain is presented first here in the text, but not first in table 2. I would suggest consistency in presentation order makes the results more reader-friendly.

We thank the reviewer for spotting this. We agree and have now changed the order for a more coherent and consistent presentation of results. 

Figures - Is it possible to amend the y-axes labels on the figures to include the weight loss method? For example, Figure 1A "gradual dieting frequency (%)". I feel this would make the figures easier to read without having to go back and forth to the figure legends.

We thank the reviewer for this suggestion, and we agree. This has now been changed on the Y-axis for a better presentation of graphs in “Fig 1.tiff”, “Fig 2.tiff” and “Fig 3.tiff”

Discussion

Line 82 - Argument regarding inexperience places athletes at greater risk of DE. Could this argument be expanded? Why would inexperience increase risk for DE/ED? Is there some protective effect in remaining in the sport, longevity helps alleviate symptoms/risk? Perhaps the lower risk in more experienced participants suggests those at risk do not remain in the sport? A similar argument is briefly presented regarding muscle dysmorphia symptoms in bodybuilders in Mitchell et al (2017) https://hes32-ctp.trendmicro.com:443/wis/clicktime/v1/query?url=http%3a%2f%2fdx.doi.org%2f10.1016%2fj.bodyim.2017.04.003&umid=fea4fa62-af1d-4bd7-b784-0a3d48fdda92&auth=e7d0e2218e6161c78f47b51039a961804123edc5-6cce90e9e58d8f2533ef7b60928430176ffc7324

Thank you for raising this valuable point. We have reflected on the reasoning behind this and expanded this discussion between the lines 378-384. In addition, we have used the Mitchell et al., (2017) reference to support the possible reason. This now reads: 

“Due to the specific demands of the Figure division (i.e., lower FM and higher FFM than the fitness division), novice athletes beginning Figure competitions may be exposed to body composition requirements that are more challenging to achieve [43]. It is also possible that women may be susceptible or already exhibit DE symptoms, as they enter physique sports, in hopes of alleviating these symptoms. As such, it is speculative whether long-term engagement in the sport provides a protective effect from ED risks in FPA.”

Lines 79-114 - following from my main comment above, this section in particular requires some thought or at least balance regarding the limitations of the EAT-26 in distinguishing pathological behaviours from inherent components of competition preparation. Line 106 regarding food restraint, perhaps food restraint is simply a requirement to continually achieve fat loss in preparation for a show.

Thank you for this really useful suggestion. We have added another sentence in the limitation paragraph between the lines 433-439 that reads: 

“Finally, although the EAT-26 questionnaire contains components that could be interpretated as pathological eating behaviours (i.e., “I engage in dieting behaviour’ and ‘Aware of the calorie content of foods that I eat’), these could well be normal behaviours that are required to compete in physique sports, rather than an DE per se. As such, distinguishing between true pathological and non-pathological behaviours eating behaviours should be carefully considered when interpretating the DE symptoms results”.

Limitations paragraph (lines 116-130) - I think worth including the limitation surrounding the EAT of distinguishing pathological from non-pathological behaviours in this particular population. Perhaps a suggestion for future research could be to validate the EAT-26 in physique athletes?

We thank the review for this comment, and we agree. We have now inserted (on line 441-442 in the future research directions of the discussion): “….., future work should i) validate the EAT-26 questionnaire in physique ….’

---

## [Decision Letter · Decision Letter 1]

18 Oct 2021

PONE-D-21-15998R1Weight loss practices and eating behaviours among female physique athletes: Acquiring the optimal body composition for competitionPLOS ONE

Dear Dr. Alwan,

Thank you for submitting your manuscript to PLOS ONE. After careful consideration, we feel that it has merit but does not fully meet PLOS ONE’s publication criteria as it currently stands. Therefore, we invite you to submit a revised version of the manuscript that addresses the points raised during the review process.

Specifically, please address the minor revisions as described by reviewer #1. 

We look forward to receiving your revised manuscript.

Kind regards,

Cherilyn N. McLester, PhD

Academic Editor

PLOS ONE

Journal Requirements:

Additional Editor Comments (if provided):

Reviewers' comments:

Reviewer's Responses to Questions

**Comments to the Author**

1. If the authors have adequately addressed your comments raised in a previous round of review and you feel that this manuscript is now acceptable for publication, you may indicate that here to bypass the “Comments to the Author” section, enter your conflict of interest statement in the “Confidential to Editor” section, and submit your "Accept" recommendation.

Reviewer #1: (No Response)

Reviewer #2: All comments have been addressed

2. Is the manuscript technically sound, and do the data support the conclusions?

Reviewer #1: Yes

Reviewer #2: Yes

3. Has the statistical analysis been performed appropriately and rigorously? 

Reviewer #1: Yes

Reviewer #2: Yes

4. Have the authors made all data underlying the findings in their manuscript fully available?

Reviewer #1: No

Reviewer #2: Yes

5. Is the manuscript presented in an intelligible fashion and written in standard English?

Reviewer #1: Yes

Reviewer #2: Yes

6. Review Comments to the Author

Reviewer #1: Dear authors

Thank you for considering the recommendations I made in the previous review. I find the manuscript improved, and have only minor remarks:

Does «chronic gradual dieting» mean that they were always on a kcal-reduced diet, or that they gradually decreased kcal-intake during the contest dieting period? If it means the latter, I suggest to omit the word “chronic”.

Line 149-150: I suggest you also add information on the total score range for the questionnaire.

Table 2: please add similar explanation on what numbers represent, as done in table 1 (and I suggest you add this to the header/the text in the title of the table): “Values are presented as mean ± SD and include the range in brackets”.

Line 283: Laxative use is mentioned as a frequent pathogenic weight control method, however the relevant question in the EAT-26 questionnaire asks about laxatives, diet pills and diuretics in the same question. As such it would be more precise to mention all these methods in the relevant section of the text and figure 3, including in the discussion. Laxative use is previously found to be less frequently used (previous mentioned references), however diet pills and diuretics are more frequently used (the latter also being anecdotally/taken from personal insight to the sport). This is underscored by the finding you specifically mention in the discussion line 350-352.

Paragraph initiated from line 374: here shortly addressing whether the EAT-26 is valid to capture DE-risk in physique athletes is recommended (I notice this has been addressed in the “limitation” of the paper, however I find It reasonable to shortly address here too, to quickly draw attention to such a dilemma).

Paragraph from line 386: First, the explanation for discrepancies presented in line 389 needs clarification (I personally did not automatically understand what the authors meant). Second; I find the most obvious reason for the discrepancies between the studies that there are different sexes in the two studies (Pickard studied male bodybuilders, you study female physique athletes). Taking the items in the EAT-26 into consideration, it is not very typical for a male body builder to respond positive to the questions in the relevant questionnaire (they are not preoccupied of being thinner, or afraid of eating/afraid of sugar or food. Not even on when being on a diet). I understand that you find it interesting to compare to a study which have used the same questionnaire, however, I find it difficult or at least less relevant, as there are obvious reasons to why there are discrepancies in your finding (type of sport i.e. bodybuilding versus fitness sports, sexes included, a questionnaire that not very well captures the issue at quest within these type of sports).

Thank you.

Reviewer #2: Thank you to the authors for addressing all of my previous comments. Well done on the preparation of this manuscript.

7. PLOS authors have the option to publish the peer review history of their article (what does this mean?). If published, this will include your full peer review and any attached files.

Reviewer #1: No

Reviewer #2: **Yes: **Lachlan Mitchell

---

## [Author Response · Author response to Decision Letter 1]

4 Dec 2021

PONE-D-21-15998R1

Weight loss practices and eating behaviours among female physique athletes: Acquiring the optimal body composition for competition

PLOS ONE

Reviewers' comments:

Reviewer's Responses to Questions

Comments to the Author

1. If the authors have adequately addressed your comments raised in a previous round of review and you feel that this manuscript is now acceptable for publication, you may indicate that here to bypass the “Comments to the Author” section, enter your conflict of interest statement in the “Confidential to Editor” section, and submit your "Accept" recommendation.

Reviewer #1: (No Response)

Reviewer #2: All comments have been addressed

2. Is the manuscript technically sound, and do the data support the conclusions?

Reviewer #1: Yes

Reviewer #2: Yes

3. Has the statistical analysis been performed appropriately and rigorously? 

Reviewer #1: Yes

Reviewer #2: Yes

4. Have the authors made all data underlying the findings in their manuscript fully available?

Reviewer #1: No

Reviewer #2: Yes

5. Is the manuscript presented in an intelligible fashion and written in standard English?

Reviewer #1: Yes

Reviewer #2: Yes

6. Review Comments to the Author

Reviewer #1: Dear authors

Thank you for considering the recommendations I made in the previous review. I find the manuscript improved, and have only minor remarks:

We thank this reviewer for their extremely thorough review and are grateful for the comments that have improved this manuscript. We are pleased to hear the reviewer feels that the manuscript has improved, and we have now attempted to address further comments in hopes the reviewer now considers the revised manuscript worthy of publication. 

Does «chronic gradual dieting» mean that they were always on a kcal-reduced diet, or that they gradually decreased kcal-intake during the contest dieting period? If it means the latter, I suggest to omit the word “chronic”.

We appreciate the reviewer’s line of thinking here. It is indeed the latter. We decided to omit “chronic” from weight loss practices throughout the manuscript. 

Line 149-150: I suggest you also add information on the total score range for the questionnaire.

We thank the reviewer for this suggestion. Unfortunately, we do not have this information as it is not provided in the original paper by Artioli et al., (2010). 

RWLS is based on scores from weight and diet history and the weight loss behaviours questions. As an example, in the weight and diet history questions, athletes will receive 1 point per kg of weight they lose, while on the weight behaviours questions, answers will be scored between 0-4. As such, it is very difficult to provide a precise range. However, we have reported: “The RWLQ scoring system indicates the higher the score obtained, the more aggressive the weight loss behaviours” on line 149-150. 

Table 2: please add similar explanation on what numbers represent, as done in table 1 (and I suggest you add this to the header/the text in the title of the table): “Values are presented as mean ± SD and include the range in brackets”.

We would like to thank the reviewer for bringing this to our attention. This has now been corrected. “Values are presented as mean ± SD and include the range in brackets” has been added in the title of the table 2 on line 221. We have also replicated this explanation in table title of table 1 on line 196-197 for consistency. 

Line 283: Laxative use is mentioned as a frequent pathogenic weight control method, however the relevant question in the EAT-26 questionnaire asks about laxatives, diet pills and diuretics in the same question. As such it would be more precise to mention all these methods in the relevant section of the text and figure 3, including in the discussion. Laxative use is previously found to be less frequently used (previous mentioned references), however diet pills and diuretics are more frequently used (the latter also being anecdotally/taken from personal insight to the sport). This is underscored by the finding you specifically mention in the discussion line 350-352.

Thank you for noting our error in reporting. This has now been corrected in the results, discussion, figure 3 title, and the y-axis on the figure 3. 

For example, this has been corrected on line 282-283: “For the combined group, 42.4% used two out of three PWCM (laxatives, diet pills and diuretics, and binge eating), while 13.3% used all methods to manage their weight. Laxatives, diet pills and diuretics….. “

Paragraph initiated from line 374: here shortly addressing whether the EAT-26 is valid to capture DE-risk in physique athletes is recommended (I notice this has been addressed in the “limitation” of the paper, however I find It reasonable to shortly address here too, to quickly draw attention to such a dilemma).

We thank the reviewer for this comment and agree with this point. We have included: “Although, EAT-26 questionnaire has been validated in other athletic groups, further work to validate it in this specific population is necessary” on line 389-391 to acknowledge this issue. 

We have now also included this additional detail in the methods section of the study design on line 120 where we directly talk about using the EAT-26 questionnaire. 

Paragraph from line 386: First, the explanation for discrepancies presented in line 389 needs clarification (I personally did not automatically understand what the authors meant). 

Second; I find the most obvious reason for the discrepancies between the studies that there are different sexes in the two studies (Pickard studied male bodybuilders, you study female physique athletes). Taking the items in the EAT-26 into consideration, it is not very typical for a male body builder to respond positive to the questions in the relevant questionnaire (they are not preoccupied of being thinner, or afraid of eating/afraid of sugar or food. Not even on when being on a diet). I understand that you find it interesting to compare to a study which have used the same questionnaire, however, I find it difficult or at least less relevant, as there are obvious reasons to why there are discrepancies in your finding (type of sport i.e. bodybuilding versus fitness sports, sexes included, a questionnaire that not very well captures the issue at quest within these type of sports).

We thank the reviewer for raising this point. We agree that a better study could be used to make comparisons to here. We have now removed these sentences from the manuscript and have included a recent study on line 376 (Money-Taylor et al., 2021) which recruited female physique athletes and used a similar screening tool to our study. 

Thank you.

Reviewer #2: Thank you to the authors for addressing all of my previous comments. Well done on the preparation of this manuscript.

We thank this reviewer for their thorough and constructive review and are grateful for the comments that have improved this manuscript.

---

## [Decision Letter · Decision Letter 2]

28 Dec 2021

Weight loss practices and eating behaviours among female physique athletes: Acquiring the optimal body composition for competition

PONE-D-21-15998R2

Dear Dr. Alwan,

We’re pleased to inform you that your manuscript has been judged scientifically suitable for publication and will be formally accepted for publication once it meets all outstanding technical requirements.

Kind regards,

Cherilyn N. McLester, PhD

Academic Editor

PLOS ONE

Additional Editor Comments (optional):

Reviewers' comments:

Reviewer's Responses to Questions

**Comments to the Author**

1. If the authors have adequately addressed your comments raised in a previous round of review and you feel that this manuscript is now acceptable for publication, you may indicate that here to bypass the “Comments to the Author” section, enter your conflict of interest statement in the “Confidential to Editor” section, and submit your "Accept" recommendation.

Reviewer #1: All comments have been addressed

2. Is the manuscript technically sound, and do the data support the conclusions?

Reviewer #1: Yes

3. Has the statistical analysis been performed appropriately and rigorously? 

Reviewer #1: Yes

4. Have the authors made all data underlying the findings in their manuscript fully available?

Reviewer #1: Yes

5. Is the manuscript presented in an intelligible fashion and written in standard English?

Reviewer #1: Yes

6. Review Comments to the Author

Reviewer #1: Thank you for the feedback on my last comments. I find the paper much improved, and it brings attention to important issues for this specific, increasingly popular sport. I have no further important comments, and recommend publication of the manuscript.

The last minor comment from me does not need to be followed up on by replies, but could be considered for additional change.

In the last review, I asked for the total score range of the “Rapid Weight loss questionnaire”, and the authors responded with a thorough explanation. Nevertheless, it would be helpful information to add comments on the possible total range for the weight behaviour subscales, as this will help us evaluate the results obtained from this questionnaire. Without this information, I would be not that intuitive to know if a total score of 22 is “knocking out the scale” or comes with medium intensity (With the information provided; if the questionnaire consists of 21 items, and a score range from 0-5 (running from 0 to 4, line 149), I’d assume there would be a total score possible of 84 points?)

Please check line 376: cut-off for what instrument? (I assume it is the EAT-questionnaire)

Line 406-407; swop the numbers; 28% had previously and 6% had currently been diagnosed with ED.

7. PLOS authors have the option to publish the peer review history of their article (what does this mean?). If published, this will include your full peer review and any attached files.

Reviewer #1: No

---

## [Editor Report · Acceptance letter]

3 Jan 2022

PONE-D-21-15998R2 

Weight loss practices and eating behaviours among female physique athletes: Acquiring the optimal body composition for competition 

Dear Dr. Alwan:

I'm pleased to inform you that your manuscript has been deemed suitable for publication in PLOS ONE. Congratulations! Your manuscript is now with our production department. 

Kind regards, 

on behalf of

Dr. Cherilyn N. McLester 

Academic Editor

PLOS ONE